



# Effects of Coupling a Stochastic Convective Parameterization with Zhang-McFarlane Scheme on Precipitation Simulation in the DOE E3SMv1 Atmosphere Model

Yong Wang[1], Guang J. Zhang[2*], Shaocheng Xie[3], Wuyin Lin[4], George C. Craig[5], Qi Tang[3], Hsi-Yen Ma[3]

[1]Ministry of Education Key Laboratory for Earth System Modeling & Department of Earth System Science, Tsinghua University, Beijing, 100084 China

[2]Scripps Institution of Oceanography, La Jolla, CA, USA

[3]Lawrence Livermore National Laboratory, CA, USA

[4]Brookhaven National Laboratory, Upton, NY, USA

[5]Meteorologisches Institut, Ludwig-Maximilians-Universität, Munich, Germany

Submitted to *GMD*

July 28, 2020

*Corresponding author:* Guang J. Zhang (gzhang@ucsd.edu)





**Abstract.** A stochastic deep convection parameterization is implemented into the U.S. Department of Energy (DOE) Energy Exascale Earth System Model (E3SM) Atmosphere Model version 1 (EAMv1). This study evaluates its performance on the precipitation simulation. Compared to the default model, the probability distribution function (PDF) of rainfall intensity in the new simulation is greatly improved. Especially, the well-known problem of "too much light rain and too little heavy rain" is alleviated over the tropics. As a result, the contribution from different rain rates to the total precipitation amount is shifted toward heavier rain. The less frequent occurrence of convection contributes to the suppressed light rain, while both more intense large-scale and convective precipitation contribute to the enhanced heavy total rain. The synoptic and intraseasonal variabilities of precipitation are enhanced as well to be closer to observations. A sensitivity of the rainfall intensity PDF to the model vertical resolution is identified and explained in terms of the relationships between convective precipitation and convective available potential energy (CAPE) and between large-scale precipitation and resolved-scale upward moisture flux. The annual mean precipitation is largely unchanged with the use of the stochastic scheme except over the tropical western Pacific, where a moderate increase in precipitation represents a slight improvement. The responses of precipitation and its extremes to climate warming are similar with or without the stochastic deep convection scheme.





# 1. Introduction

Precipitation plays a vital role in the Earth's climate: the latent heat released during precipitation formation is a major energy source that drives the atmospheric circulation, and the precipitation is an important part of the Earth's hydrological cycle. The accurate simulation of precipitation in global climate models (GCMs) is of great scientific and societal interest. However, GCMs used for current climate simulation and future projections suffer from many biases in the global distribution, frequency and intensity of simulated precipitation (Dai, 2006), which have negatively impacted the model's fidelity. Rainfall in nature is tightly associated with many complex dynamic and physical processes in the atmosphere, including large-scale circulation, convection, cloud microphysics, and planetary boundary layer (PBL) processes. The deficiencies in representing these processes in GCMs are prime culprits for errors in simulated rainfall (Watson et al., 2017).

Among the physical processes in GCMs, the parameterization of convection is responsible for some well-known biases: the double Intertropical Convergence Zone (Zhang and Wang 2006; Zhang et al., 2019), too weak synoptic and intraseasonal variabilities in the tropics (Zhang and Mu, 2005a; Watson et al., 2017), the wrong diurnal cycle of rainfall (Xie et al., 2019), "too much light rain and too little heavy rain" (Dai, 2006; Zhang and Mu, 2005b; O'Gorman and Schneider, 2009), to name a few. The conventional deterministic convective parameterization in GCMs represents the ensemble effects of subgrid-scale convective clouds in a model grid box on resolved scale variables. However, in reality, a given grid-scale state may lead to different realizations of subgrid-scale convection (Davies et al., 2013; Peters et al., 2013) rather than to a single "ensemble-mean" response. For instance, two model grid boxes, both in a similar convective-equilibrium state, can have different numbers and/or sizes of convective clouds due to stochasticity (Cohen and Craig, 2006). This stochasticity will appear more frequently as the model grid-box size becomes smaller (Jones and Randall, 2011). Not including stochasticity in convective schemes has been suggested to be at least partly responsible for the weak intraseasonal variability and "too much light rain and too little heavy rain" in GCMs (Lin and Neelin 2000, Wang et al., 2016; Watson et al., 2017; Peters et al., 2017).

As suggested in Palmer (2001, 2012), more realistic statistics of the impacts of subgrid convective clouds should be derived by simulating them as random samples from probability distributions conditioned on the grid-scale state, so that the influences of different individual realizations are introduced in the convection parameterization. In this regard, much effort in the





past two decades has been made to develop stochastic convection schemes (e.g., Lin and Neelin, 2000, 2002; Plant and Craig, 2008; Khouider et al., 2010; Sakradzija et al., 2015). Among these schemes, Plant and Craig (2008) (PC08 hereafter) developed a stochastic deep convection parameterization under a framework based on statistical mechanics (Cohen and Craig, 2006; Craig and Cohen, 2006) for noninteracting convective clouds in statistical equilibrium using cloud-resolving model (CRM) simulations. This scheme was applied to numerical weather prediction (NWP) models and to a GCM in an aquaplanet setting, resulting in some substantial improvements in precipitation simulation (Groenemeijer and Craig, 2012; Keane et al., 2014, 2016).

Wang et al. (2016) incorporated the PC08 stochastic deep convection scheme into the Zhang-McFarlane (ZM) deterministic deep convection scheme (Zhang and McFarlane, 1995) in the National Center for Atmospheric Research (NCAR) Community Atmosphere Model version 5 (CAM5). They found that the introduction of the stochastic scheme improved the simulation of precipitation intensity and intraseasonal variability over the tropics in CAM5 (Wang and Zhang 2016; Wang et al., 2017).

In this study, we implement the PC08 stochastic deep convection parameterization scheme into the DOE Energy Exascale Earth System Model (E3SM) (Golaz et al. 2019) Atmosphere Model version 1 (EAMv1) (Rasch et al. 2019; Xie et al. 2018) and examine its effect on precipitation simulation. The EAMv1 is branched out from CAM5 and thus it inherits many model deficiencies from CAM5 as well. Many modifications in physics parameterizations have been made compared to CAM5 (Rasch et al. 2019; Xie et al. 2018). However, some model biases such as weak precipitation intensity persist (Xie et al. 2019). Thus, besides the precipitation metrics explored in our previous studies (Wang et al. 2016, 2017; Wang and Zhang 2016), this study will evaluate precipitation simulation with more systematical metrics. In addition, the responses of precipitation and its extremes to climate warming with the stochastic deep convection scheme will be investigated.

The organization of the paper is as follows. Section 2 presents parameterization, model, experimental design, and evaluation data. Section 3 describes results, including variability, frequency, intensity, amounts, duration, mean state, and responses of precipitation and its extremes to climate warming. The sensitivity of the rainfall intensity pdf to vertical resolution and underlying mechanisms are also presented in this section. Summary is given in section 4.

## 2. Parameterization, model, experimental design and evaluation data



## 2.1. Stochastic deep convection parameterization

The stochastic convective parameterization scheme of PC08 is modified for climate models
when incorporating into the ZM deterministic deep convection scheme. In the PC08 scheme, the
probability of launching one convective cloud is given by:

$$p_{d\bar{n}(m)}(n=1) = \frac{1}{\langle m \rangle} e^{-m/\langle m \rangle} \langle N \rangle dm \quad (1)$$

where $d\bar{n}(m)$ denotes the average number of clouds with mass flux between $m$ and $m+\mathrm{d}m$, $<m>$
is the ensemble mean mass flux of a cloud, and $<N>$ is the ensemble mean number of convective
clouds in a given GCM grid box ($<N>=<M>/<m>$, with $<M>$ the ensemble mean total cloud mass
flux given by the closure in the ZM deterministic parameterization). For each mass flux bin,
whether to launch a cloud is determined by comparing the probability from Eq. (1) with a random
number uniformly generated between zero and one which, unlike the update frequency of once a
day in Wang et al. (2016), is updated every 3 days in consideration of computational resources due
to finer vertical and horizontal resolutions in the EAMv1 (see section 2.2). For the same reason,
the spatial averaging of input quantities (i.e., vertical profiles of temperature and moisture) to the
stochastic scheme over neighboring grid points used in the original design of PC08 is not
performed because it leads to an excessive communication load. One can argue that at a horizontal
model resolution of about 110 km in EAMv1, convective quasi-equilibrium approximately holds
over some timescale although at individual model timestep it does not. Thus, although spatial
averaging is not applied, the temporal trailing averaging over 3 h at each time step is retained in
the scheme. Other modifications to the PC08 scheme for incorporation into the ZM scheme in
climate models (Wang et al. 2016) are retained. These include:
1) The temporally averaged quantities are used to calculate the ensemble mean cloud mass
flux ($<M>$), which is determined by the ZM scheme. The unsmoothed grid point quantities are still
used in the trigger function and the cloud model.
2) The root mean squared cloud radius information originally used in PC08 is not needed in
our implementation because the ZM scheme does not use cloud radius.
3) The ensemble mean mass flux of a cloud $<m>$ is set to $1 \times 10^7$ kg s$^{-1}$ following
Groenemeijer and Craig (2012).
4) The cloud life cycle effect with a factor $\mathrm{d}t/T$ ($\mathrm{d}t$ is the model time step and $T$ is the constant
lifetime parameter) in PC08 is not taken into account because the ZM deterministic
parameterization does not consider the life cycle of convection.



5) The mass fluxes from all clouds in a GCM grid box generated from eq. (1) are rescaled by a factor $<N>$ to account for the fact that there can be many clouds in a GCM grid box.

## 2.2.   EAMv1 model

The standard configuration of the DOE EAMv1 uses a spectral element dynamical core at 110-km horizontal resolution on a cubed sphere geometry and a vertical resolution of 72 layers from the surface to 60 km (10 Pa) (Rasch et al. 2019, Xie et al. 2018). The treatment of PBL turbulence, shallow convection, and cloud macrophysics are unified with a simplified third-order turbulence closure parameterization CLUBB (Cloud Layers Unified by Binormals, Golaz et al., 2002; Larson and Golaz, 2005). The deep convection is represented by the ZM scheme. The Morrison and Gettelman (2008) (MG) microphysics scheme is updated to MG2 (Gettelman et al., 2015) with the prediction of rain and changes to ice nucleation and ice microphysics (Wang et al., 2014). A four-mode version of the modal aerosol module (MAM4) (Liu et a., 2016) is used with improvements to aerosol resuspension, aerosol nucleation, scavenging, convective transport and sea spray emissions for including the contribution of marine ecosystems to organic matter (Rasch et al., 2019). A linearized ozone chemistry module (Hsu and Prather, 2009; McLinden et al., 2000) is used to represent stratospheric ozone and its radiative impacts in the stratosphere. Other modifications for model tuning are provided in detail in Xie et al. (2018).

## 2.3.   Experimental design

Six Atmospheric Model Intercomparison Project (AMIP) type simulations are conducted. Four 6-year simulations are forced by prescribed, seasonally varying climatological present-day sea surface temperatures (SSTs) and sea ice extent, recycled yearly (Stone et al., 2018): two with the default deterministic ZM scheme but having 72 and 30 vertical levels respectively (referred to as EAMv1 and EAMv1-30L) and the other two with the stochastic deep convection scheme (referred to as STOCH and STOCH-30L). The simulations with 30 vertical levels are conducted to facilitate the comparison with Wang et al. (2016), in which the vertical resolution of CAM5 is 30 levels (see section 3.3). To explore the responses of precipitation and its extremes to climate warming, similar to EAMv1 and STOCH runs, two 3-year simulations in a warmer climate are conducted, in which a composite SST warming pattern derived from the Coupled Model Intercomparison Project Phase 3 (CMIP3) coupled models (referred to as EAMv1-4K and STOCH-4K respectively) is imposed for the boundary condition of the atmosphere. Following





Webb et al. (2017), it is a normalized multi-model mean of the sea surface temperature response
pattern from 13 CMIP3 atmosphere-ocean general circulation models, representing the change of
SST between years 0-20 and 140-160, the time of $CO_2$ quadrupling in the 1% runs. Before
calculating the multi-model ensemble mean, the SST response of each model was divided by its
global mean and multiplied by 4K. This guarantees that the pattern information from all models is
weighted equally and that the global mean SST forcing is +4K warming. The first year in all
simulations is discarded as a spin-up. Information for all experiments is summarized in Table 1.

**2.4.   Evaluation data**

For model evaluation, the following datasets are used: The Clouds and the Earth's Radiant
Energy System Energy Balanced and Filled (CERES-EBAF) (Loeb et al., 2009) for evaluation of
shortwave and longwave cloud radiative forcing; the Interim European Centre for Medium-Range
Weather Forecasts Re-Analysis (ERAI) (Simmons et al.,2007) for sea level pressure, zonal wind,
relative humidity, specific humidity, and temperature; the European Remote Sensing Satellite
Scatterometer (ERS) (Bentamy et al., 1999) for surface wind stress; and the Willmott-Matsuura
(Willmott) (Willmott & Matsuura, 1995) data for land surface air temperature.

The rainfall mean state is evaluated against the Global Precipitation Climatology Project
(GPCP) monthly product (version 2.1) at a resolution of 2.5º (Adler et al., 2003; Huffman et al.,
2009) while a daily estimate of GPCP version 1.2 at 1º horizontal resolution (GPCP 1DD)
(Huffman et al., 2001, 2012) is used for evaluation of precipitation amount distribution. In addition
to GPCP, the Xie-Arkin pentad observations at 2.5º resolution (Xie and Arkin, 1996) and the
Tropical Rainfall Measuring Mission 3B42 version 7 (TRMM) daily observations at a resolution
of 0.25º over (50ºS, 50ºN) (Huffman et al., 2007) are applied to evaluate the precipitation variance,
while the latter is also used in the PDF of rainfall intensity and the rainfall amount distribution.
For the rainfall duration evaluation, the TRMM 3B42 v7 3-hourly data is used. To make the
comparison consistent between observations and model simulations, the model data with the same
output frequency to that in the corresponding observations/reanalysis data are used and all
observations/reanalysis data are regridded to the same 1º lat-lon grids as EAMv1.

**3.    Results**
**3.1.   Intraseasonal and synoptic variability**

The simulated variability of precipitation is an important aspect of model performance. Here





we focus on intraseasonal and synoptic-scale variability. The intraseasonal variability associated
with Madden-Julian oscillations (MJO) is problematic in many GCMs (Jiang et al. 2015; Zhang
and Mu 2005). Figure 1 shows the tropical distribution of the 20-80 day intraseasonal variance for
the total precipitation in observations and simulations. The variance is obtained with a Lanczos
band-pass filter at each grid point. Both Xie-Arkin and TRMM observations show large variance
in the Indian Ocean and western Pacific as well as in the ITCZ and the South Pacific Convergence
Zone (SPCZ) regions. The intraseasonal variance in EAMv1 is much weaker, as in many other
GCMs. Similar to the results in Wang et al. (2016), the STOCH run with the stochastic deep
convection scheme has a significantly enhanced intraseasonal variance in these regions, making it
much more comparable to observations.
Besides the intraseasonal variance, the synoptic variance (2-9 day Lanczos band pass-filtered
rainfall anomalies) is also investigated (Fig. 2). The synoptic-scale variance corresponds to
weather activities. In Fig. 2 only TRMM observations are shown to evaluate simulations because
the Xie-Arkin observations are pentad data. In TRMM, the geographical distribution of the
synoptic variance is similar to that of the intraseasonal variance, but with larger amplitudes because
synoptic-scale activities contain much more energy than intraseasonal disturbances. Similar to the
intraseasonal variance, the synoptic variance in the EAMv1 run is also much weaker than that in
observations. The synoptic-scale variance in the STOCH run is about twice as strong as in EAMv1
although it is still underestimated compared to TRMM observations. This result is consistent with
Goswami et al. (2017), which reported enhanced intraseasonal and synoptic variability of
precipitation in the National Centers for Environmental Predictions (NCEP) Climate Forecast
System version 2 (CFSv2) using a stochastic multicloud model parameterization.

### 3.2. Rainfall frequency, intensity, amount and duration

Wang et al. (2016) showed that the most significant improvement with the use of the
stochastic deep convection scheme in CAM5 was in the simulated PDF of rainfall intensity over
the tropics, which became very close to TRMM observations. Since there are many modifications
in model configuration and physics parameterizations from CAM5 to EAMv1 (Rasch et al. 2019),
such as a finer vertical resolution, an updated microphysics parameterization (MG2), and the use
of CLUBB in place of separate shallow convection and planetary boundary layer turbulence
parameterizations, it is not clear whether a similar degree of improvement in precipitation intensity
PDF can be achieved with a similar stochastic convection scheme. Using an equal-interval rainfall

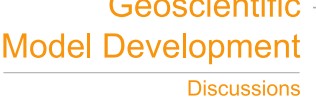

intensity bin of 0.5 mm d$^{-1}$ from 0 to 200 mm d$^{-1}$, Fig. 3 shows the frequencies of the total
precipitation intensity over the tropics (20°S-20°N) from TRMM, EAMv1 and STOCH,
respectively. Also shown are the PDFs of large-scale and convective precipitation intensity. As
seen in Fig. 3a, the stochastic convection parameterization in the STOCH run greatly mitigates the
bias of "too much light rain and too little heavy rain", showing a decrease of the frequency of
rainfall intensity between 1 and 10 mm d$^{-1}$ and an increase of that of rainfall intensity larger than
20 mm d$^{-1}$ compared to the EAMv1 run. Xie et al. (2019) indicated that the "too much light rain"
in EAMv1 was a result of too frequent convection. Consistent with this notion, Fig. 3b shows that
the reduction of the light rain frequency is entirely from convective precipitation. On the other
hand, the increase of intense precipitation frequency is from both convective and large-scale
precipitation.

To understand why the use of stochastic convection scheme decreases the frequency of light

rain and increases the frequency of heavy rain, we conducted an additional simulation. In the
simulation, the setup is identical to the STOCH run except that the ZM scheme is called a second
time at each time step, with input (temperature, moisture, etc.) identical to that for the stochastic
scheme. However, the output is used for diagnostic purposes only and does not participate in model
integration. It is found that (figure not shown) two factors contribute to the decreased frequency
of light rain and increased frequency of heavy rain. First, for a given ensemble mean convective
mass flux (from the ZM scheme) the probability for cloud generation following the Poisson
distribution for a realization in the stochastic scheme can produce more intense precipitation than
obtained by the ZM scheme. Second, the probability distribution results in less frequent convection
in general. This allows the buildup of the atmospheric instability (also see Fig. 9 below in section
3.3), which also leads to heavier convective rainfall (even with ZM scheme alone without
considering the stochastic part) as well as more large-scale condensation. However, we note that
the increase of the frequency in rainfall intensity ranges from 60 to 140 mm d$^{-1}$ in the STOCH run
is not as much as that in Wang et al. (2016) for CAM5. This will be elucidated through sensitivity
experiments in the next subsection.

The frequencies of total precipitation intensity over selected regions also show qualitatively

similar degree of improvement. Fig. 4 shows six regions during their convectively active seasons:
Amazonia, tropical western Pacific, India for June-September, Maritime Continent, Southern
Great Plains (SGP) for May-August and eastern China for June-August in TRMM, EAMv1 and
STOCH, respectively. In all tropical regions, the EAMv1 simulation overestimates the occurrence





frequency for precipitation intensities less than 20 mm day$^{-1}$ and underestimates it for precipitation
intensities greater than 20 mm day$^{-1}$, similar to the distribution for the entire tropics. In STOCH,
the performance in the pdf over Amazonia and Maritime Continent is better than the pdf over the
entire tropics. Although the biases of "too much light rain" over India and tropical western Pacific
are alleviated by the stochastic deep convection scheme, the bias of "too little heavy rain" remains,
particularly over India where large-scale monsoonal dynamics regulate heavy convective rain
(Wang et al., 2018). For the two midlatitude convection regions (SGP and eastern China), although
there is also noticeable improvement across the precipitation intensity spectrum, it is less
significant compared to other regions, possibly because convection in midlatitude land regions is
not as prevalent as in the tropics.
Figure 5 shows the geographical distributions of precipitation frequency for all precipitation,
for precipitation intensities less than 20 mm d$^{-1}$, and more than 20 mm d$^{-1}$, respectively, over the
tropics in observations and simulations (days with precipitation intensity less than 1 mm d$^{-1}$ are
considered non-precipitating and thus excluded). In TRMM, the occurrence frequency of rainy
days ranges from 30 to 70% with the most frequent rain along the ITCZ, the SPCZ and in the
Indian Ocean, where the EAMv1 run has as high a frequency as 80-90%, with up to 30% positive
biases. In contrast, the STOCH run reduces the frequency to 50-70% although it is still
overestimated. When the total precipitation is broken down into precipitation rates less than 20
mm d$^{-1}$ and precipitation rate above 20 mm d$^{-1}$, in both observations and simulations the
geographical distribution of the rainy days is dominated by that of days with precipitation intensity
less than 20 mm d$^{-1}$. In comparison with observations, again, the STOCH run reduces the positive
bias of the frequency of precipitation intensity less than 20 mm d$^{-1}$ in the EAMv1 run by up to
20%. For precipitation intensities greater than 20 mm d$^{-1}$, the EAMv1 run underestimates their
frequency compared to the TRMM observations. On the other hand, the frequency of occurrence
in the STOCH run is comparable to the TRMM observations.
Another metric for the precipitation pdf is the contribution of precipitation within a given
intensity bin to the total precipitation amount. It combines the information of precipitation
frequency distribution and precipitation intensity. While drizzle occurs much more frequently than
the more intense rain events, it may not contribute much to the total precipitation amount.
Following the approach of Kooperman et al. (2016, 2018), we divide the precipitation rate ranging
from 0.1 to 1000 mm d$^{-1}$ into equal bin intervals on a logarithmic scale, with a bin width of
$\Delta \ln(R) = \Delta R/R = 0.1$. If the frequency of rainfall rates falling into the $i$th bin is denoted $f_i$, then



$f_i = n_i/N_t$, where $N_t$ is the total number of days, $n_i$ is the number of days with rainfall rates
falling into the $i$th bin. The mean precipitation rate in the $i$th bin is then:
$$R_i = \frac{1}{n_i}\sum_{j=1}^{n_i} r_j, \quad (2)$$

where $r_j$ is an individual precipitation rate within the $i$th bin. Thus, the contribution to the total
precipitation amount from the $i$th bin per unit bin width is given by:
$$P_i = \frac{f_i R_i}{\Delta \ln(R)} = \frac{1}{\Delta \ln(R)}\frac{1}{N_t}\sum_{j=1}^{n_i} r_j \quad (3)$$

$P_i$ has the units of mm d$^{-1}$. The total precipitation amount is then given by:
$$P = \sum_i P_i \,\Delta \ln(R) = \sum_i f_i\, R_i \quad (4)$$

Accordingly, the amount distributions for total ($P^T$), convective ($P^C$) and large-scale ($P^L$) rainfall
are given by:
$$P_i^T = \frac{1}{\Delta \ln(R)}\frac{1}{N_t}\sum_{j=1}^{n_i} r_j^T \quad (5)$$

$$P_i^C = \frac{1}{\Delta \ln(R)}\frac{1}{N_t}\sum_{j=1}^{n_i} r_j^C \quad (6)$$

$$P_i^L = \frac{1}{\Delta \ln(R)}\frac{1}{N_t}\sum_{j=1}^{n_i} r_j^L \quad (7)$$

where $r^T$, $r^C$ and $r^L$ are the total, convective and large-scale rain rates.
Figure 6a shows the contribution to the total rainfall amount from each rainfall rate on a
logarithmic scale for GPCP 1DD, TRMM, and the two simulations, respectively, over the tropics.
The TRMM observations have larger contributions from intense rainfall rates than GPCP 1DD,
with the peak contribution rainfall rate of 28 mm d$^{-1}$, higher than the value of 22 mm d$^{-1}$ in GPCP
1DD. The EAMv1 run produces a much smaller peak contribution rainfall rate (15 mm d$^{-1}$) than
the two observations while the STOCH run simulates it realistically (23 mm d$^{-1}$), falling in between
the two observations. Note that precipitation from intensities less than 1 mm d$^{-1}$ contributes about
0.05 mm d$^{-1}$ or less to the tropical mean total precipitation, thus justifying treating it as non-
precipitating in Fig. 5. Fig. 6b shows the convective and large-scale contributions to the simulated
total precipitation from EAMv1 and STOCH, respectively. The large-scale precipitation shows
very similar contribution distributions in the two simulations, except for the largest rain rates which
make only a small contribution to the total. For the most part, large-scale precipitation is not
affected by how convection is treated in the model, with both simulations having a maximum
contribution near 22 mm d$^{-1}$. On the other hand, the convective contribution is very different



between the two simulations. Similar to the total precipitation, the peak contribution to convective
precipitation is at a much smaller rainfall rate in EAMv1 than in STOCH.

Besides precipitation frequency and intensity, another important higher order statistic of

precipitation is the duration of precipitation events; it measures the intermittency of precipitation
(Trenberth et al. 2017). Using 3-hourly data, we calculate the duration of rainfall events as
continuous number of hours of precipitation exceeding a threshold value of 1 mm $d^{-1}$. Figure 7
shows the frequency of precipitation events for different durations over the tropics. 80% of TRMM
observed precipitation events lasts for 3 hours or less, 18% lasts for 6 hours and 2% lasts for 9
hours. In contrast, both EAMv1 and STOCH produce very small proportions (~15%) of
precipitation events that last for 3 hours or less. The frequency of precipitation events lasting 9
hours or longer is extremely overestimated in the model simulations, with some lasting for as long
as 21 hours. This suggests that convection in the model lacks the observed intermittency (Trenberth
et al. 2017) and the use of the stochastic convection scheme does not improve this aspect of the
simulated convection.

**3.3 Sensitivity of rainfall intensity PDF to vertical resolution**

A significant modification among several changes in EAMv1 from CAM5 is a much finer

vertical resolution, increasing from 30 levels in CAM5 to 72 levels in EAMv1. Within the PBL
alone EAMv1 has 17 layers compared to 5 layers in CAM5, and the thickness of approximately
20 m for the lowest model layer in EAMv1 is much thinner than that in CAM5, which is 100 m
(Xie et al., 2018). The increased resolution in the PBL in EAMv1 will likely affect the convection
behavior through PBL-convection interactions. In Fig. 3 we showed that the precipitation intensity
pdf is significantly improved with the introduction of the stochastic convection scheme. However,
the improvement was not as striking as that shown in Wang et al. (2016) for CAM5. We suspect
that this is primarily due to the enhanced vertical resolution in EAMv1 rather than other changes
in model physics parameterizations, tunings, or the model dynamic core. To confirm this, EAMv1-
30L and STOCH-30L runs with a vertical resolution of 30 layers are conducted and compared with
the EAMv1 and STOCH runs with the default 72 vertical layers. As seen in Figure 8, when
switching to a configuration of 30 vertical layers, the performance of the STOCH-30L run is very
similar to that in CAM5 (Wang et al., 2016). The frequency distribution of rainfall intensity
between 60 and 140 mm $d^{-1}$ almost falls on top of that in TRMM. The PDF of rain intensity in the
EAMv1-30L run is also closer to TRMM observations compared to the EAMv1 run (Fig. 8a). For





EAMv1, both convective and large-scale precipitation becomes more intense in the 30-level
configuration. In contrast, the frequency of more intense convective precipitation in STOCH-30L
is increased while that of more intense large-scale precipitation is decreased (Fig. 8b&c), similar
to the dependence of precipitation pdf on horizontal resolution documented in previous studies,
which showed that refining the horizontal resolution should result in more large-scale precipitation
and less convective precipitation (e.g., O'Brien et al., 2016).
The causes of sensitivity of convective and large-scale precipitation to vertical resolution are
further examined below. In the ZM convection scheme, the amount of convection is linked to
convective available potential energy (CAPE). Thus, in Figure 9 we present the joint PDF of
convective precipitation and CAPE over the tropics in the four simulations. Note that all parameter
settings are identical between EAMv1 and EAMv1-30L except the vertical resolution. Both
EAMv1 and EAMv1-30L show an approximately linear relationship between CAPE and
convective precipitation. CAPE values are generally smaller in EAMv1-30L than in EAMv1, as
can be seen from the frequency of occurrence of both large and medium CAPE values. However,
the slope of the maximum occurrence frequency is almost twice as large in EAMv1-30L as in
EAMv1 (Fig. 9a&b), giving higher frequency of larger convective precipitation as seen in Fig. 8.
This result is puzzling to us at first. However, note that for a given precipitation rate that the model
produces, there is in general a large range of CAPE values and the CAPE values in EAMv1 are
predominantly larger than in EAMv1-30L as can be seen from the pdf distribution in Fig. 9a and
b. Compared to EAMv1, the smaller CAPE values in EAMv1-30L are caused by higher parcel
launching levels due to thicker model layers near the surface, where the most unstable air is often
found (figure not shown). There is also a bifurcation for medium to large CAPE values. This is
likely related to atmospheric moisture conditions in the atmosphere: for the same CAPE values
there is less precipitation when the atmosphere is dry, and vice versa. With the introduction of the
stochastic deep convection scheme, there are no longer an approximately linear relations between
CAPE and convective precipitation (Fig. 9c&d) in spite of the fact that the CAPE-based closure is
still used to determine the cloud base mass flux (presumably ensemble mean). This is surprising;
it implies that for a given convectively unstable atmospheric thermodynamic condition, the use of
the stochastic scheme often inhibits the triggering of convection, thereby allowing for the buildup
of CAPE for (the less frequently occurring) stronger convection. Similar to EAMv1, smaller
(larger) CAPE values occur more (less) frequently in STOCH-30L due to higher parcel launching
levels. Also, the small and moderate values of CAPE have larger probabilities to precipitate more





in STOCH-30L compared to STOCH.
Because large-scale precipitation is related to resolved-scale upward moisture flux $-\omega q/g$,
where $\omega$ is vertical velocity in pressure coordinate, $q$ is specific humidity and $g$ is gravitational
acceleration (O'Brien et al., 2016), Fig. 10 shows the PDFs of upward moisture flux at 850 hPa in
the simulations. In comparison with the 72-level configuration, EAMv1-30L has larger frequencies
for upward moisture fluxes larger than 20 mm d$^{-1}$ while STOCH-30L has larger frequencies for
upward moisture fluxes from 20 to 80 mm d$^{-1}$ but smaller frequencies for fluxes larger than 80 mm
d$^{-1}$. These correspond well with the changes in the PDF of large-scale precipitation from the 30-
level to the 72-level simulations in Fig. 8.

**3.4 Mean state**

So far, we have shown that the introduction of a stochastic convection scheme into the E3SM
atmospheric model can significantly improve the simulation of short-term variability and intensity
pdf of precipitation. In climate model development efforts, it is important that an improvement in
some aspects of the model does not lead to degradation of other aspects, at least not to outweigh
the improvement. Thus, it is imperative that we examine the climate mean fields as well. Fig. 11
shows the global distribution of annual mean precipitation in GPCP observations and simulations,
as well as the differences of total, convective, and large-scale precipitation between the STOCH
and EAMv1 runs. Overall, the geographical distributions of precipitation in the two simulations
are similar to those in observations, but both overestimate the tropical precipitation (Fig. 11a-c).
There is a slight increase of rainfall over the tropical western Pacific, equatorial Indian Ocean and
Africa and a slight decrease over India and Amazonia in the STOCH simulation (Fig. 11d). Most
of these changes are from convective precipitation except over equatorial Africa where the changes
are from large-scale precipitation (Fig. 11e&f).
The zonal mean of temperature and specific humidity from ERAI and the model biases are
shown in Figure 12. For temperature, EAMv1 produces mostly negative biases in the entire
troposphere over the tropics and subtropics and positive biases in the lower troposphere in high
latitudes. With the stochastic deep convection scheme used, the temperature changes in STOCH
are very minor, increasing slightly from EAMv1. In the simulation of specific humidity, there are
positive biases in the lower troposphere across all latitudes and negative biases above 900 hPa over
the tropics and subtropics in EAMv1. In comparison with EAMv1, the negative biases are
alleviated but the positive biases are increased slightly in STOCH.





The overall difference in model performance as measured by the commonly used mean
climate metrics between EAMv1 and STOCH runs is summarized in the Taylor diagram (Fig. 13).
Most metrics are comparable between the two simulations except precipitation, especially over
land where STOCH shows a larger standard deviation than both GPCP and EAMv1. In short, the
mean climate does not change much after the incorporation of the stochastic convection scheme
in EAMv1. This is practically desirable since one does not need to heavily re-tune the model, a
task that is often time-consuming and more of engineering than scientific interest.

**3.5.    Response to climate warming**

Another aspect of interest concerns the model's response to climate change. It is well known
that the estimated climate sensitivity for future climate projections is sensitive to changes in model
physics parameterizations (Golaz et al. 2019). With the stochastic deep convection
parameterization, it is necessary to check if the response of precipitation and associated extremes
to climate warming differs. As seen in Fig. 14, relative to the current climate simulations, the
geographical patterns and magnitudes of annual mean precipitation changes normalized by the
global-mean surface air warming ($\Delta T_{sa}$) in the +4K SST warming simulations (i.e., $(P_{+4k} -$
$P)/P/\Delta T_{sa}$, units: %/K) with and without the stochastic deep convection scheme are very similar,
both showing maximum increases over the ITCZ, the western Pacific and the Indian Ocean.
Pendergrass et al., (2019) found that the response of extreme precipitation to warming follows a
nonlinear relation:
$$\frac{dr_x}{dT_{sa}} = aT_{sa} \quad (8)$$

or
$$r_x = \frac{1}{2}aT_{sa}^2 + b \quad (9)$$

where $r_x$ is a rainfall extreme index (here using R95p, the total rainfall from the days with daily
rainfall intensity exceeding 95th percentile of the daily precipitation distribution), $T_{sa}$ is the
global-mean surface air temperature in a warmer world, and $a$ is the slope of $dr_x/dT_{sa}$ versus
$T_{sa}$ measuring the strength of the nonlinear response of extreme rainfall to warming. At each grid
point, $dr_x \approx \Delta r_x$ is equal to R95p in a warmer world minus that under the current climate and
normalized by the global-mean surface air warming ($dT_{sa} \approx \Delta T_{sa}$). With $T_{sa}$ in the +4K SSTs
warming simulations and the calculated $dr_x/dT_{sa}$, the global distributions of the slope, $a$
(units: %/K$^2$), with and without the stochastic deep convection scheme are displayed in Fig. 14c&d.



Although the stochastic deep convection parameterization introduces stochasticity into convection and significantly improves the underestimated frequency of intense precipitation under the current climate (Wang et al., 2017), it does not lead to a different nonlinear response of precipitation extremes in a warmer world. Increasing circulation strength as climate warms is considered to be the main driver for the nonlinear relationship between tropical precipitation extremes and global-mean surface air temperature (Pendergrass et al., 2019), and it is possible that the circulation changes with and without the stochastic deep convection scheme are similar. Relative to their respective current climate states, the responses of the EAMv1-4K and STOCH-4K runs show similar geographical distributions with comparable maximum nonlinearity over the tropical Pacific and Atlantic and the Indian Ocean which bears some resemblance to that in Pendergrass et al. (2019).

## 4. Summary

In this study, we implemented the stochastic deep convection scheme (Plant and Craig, 2008; Wang et al., 2016) into the DOE EAMv1 and investigated its impact on the simulation of precipitation. Several improvements are observed with the use of the stochastic convection scheme: (1) the weak intraseasonal and synoptic-scale variabilities in EAMv1 are enhanced to levels much closer to those in observations; (2) the "too much light rain and too little heavy rain" bias over the tropics is significantly alleviated due to less frequent occurrence of drizzling convection and more frequent occurrence of intense large-scale and convective precipitation contributing to enhanced heavy rain; (3) the simulated peak precipitation rates (the amount mode) in the precipitation amount distribution, which contribute the most to the total amount of precipitation, are larger and are in better agreement with those in TRMM and GPCP observations.

While the improvement in the simulated PDF of rainfall intensity is significant, it is less than what we had expected based on our earlier work with the NCAR CAM5 (Wang et al., 2016). Since there are many changes from CAM5 to EAMv1, including vertical resolution, model dynamic core and physics parameterizations, it is not clear which changes are related to the difference in the improvement of the simulated rainfall pdf. Two sensitivity tests were performed to elucidate this, both with a coarser vertical resolution configuration of 30 layers (i.e., EAMv1-30L and SOTC-30L) as in CAM5. The STOCH-30L run successfully reproduces the frequency distribution of rainfall intensity found by Wang et al. (2016) with an increased frequency of convective precipitation intensities between 60 and 140 mm d$^{-1}$. This increase is explained by the fact that





small and moderate values of CAPE generate more convective precipitation from the altered
relation between them compared to the 72-level configuration due to fewer model layers in the 30-
level resolution. Large-scale precipitation is also influenced by the vertical resolution, but it
behaves differently in EAMv1-30L and STOCH-30L compared with EAMv1 and STOCH
respectively because of the different response of the resolved-scale upward moisture flux at 850
hPa.
For any changes in model physics parameterization that improve some aspects of the model
performance, it is important that other aspects are not degraded. It is known in the climate modeling
community that improved intraseasonal variability is often accompanied by a degradation of the
mean state (e.g., Kim et al. 2011; Klingaman and Demott, 2020). We showed that the mean states
in tropospheric temperature, moisture as well as precipitation are not much different with or
without the use of the stochastic convection scheme, and neither are the responses of mean
precipitation and precipitation extremes to climate warming. This is encouraging and desirable for
model development efforts. However, we note that for higher horizontal resolutions (Caldwell et
al., 2019) or a regionally refined mesh version of EAMv1 (Tang et al., 2019), spatial averaging of
the input fields of the stochastic scheme would be needed to make use of convective quasi-
equilibrium over a larger domain. This could be challenging for computational efficiency and it
requires further research in the future.

**Code and data availability.** The E3SMv1 source code can be downloaded from the E3SM official
website https://e3sm.org/. The GPCP 1DD and TRMM 3B42 data are available from NASA GSFC
RSD        (https://psl.noaa.gov/data/gridded/data.gpcp.html)       and       Mirador
(http://mirador.gsfc.nasa.gov), respectively. The EAMv1 simulation output is provided in an open
repository Zenodo (http://doi.org/10.5281/zenodo.3902998).

**Author contributions.** GJZ conceived the idea. YW developed the model code. YW and WYL
conducted the model simulations. YW performed the analysis. YW and GJZ interpreted the results
and wrote the paper. All authors participated in the revision and editing of the paper.

**Acknowledgements:** This work is supported by the National Key Research and Development
Program of China Grants 2017YFA0604000, and the National Natural Science Foundation of
China Grants 41975126 and 41605074. GJZ is supported by the Department of Energy, Office of



Science, Biological and Environmental Research Program (BER), under Award Numbers DE-
SC0019373 and DE-SC0016504. GCC is supported by subproject A1 of the Transregional
Collaborative Research Center SFB / TRR 165 "Waves to Weather" (www.wavestoweather.de)
funded by the German Research Foundation (DFG). Work at LLNL was performed under the
auspices of the U.S. DOE by Lawrence Livermore National Laboratory under contract No. DE-
AC52-07NA27344. SX and QT are supported by the DOE Energy Exascale Earth System Model
(E3SM) project and HYM is funded by the DOE Regional and Global Model Analysis program
area (RGMA) and ASR's Cloud-Associated Parameterizations Testbed (CAPT) project. This
research used resources of the National Energy Research Scientific Computing Center, a DOE
Office of Science User Facility supported by the Office of Science of the U.S. DOE under Contract
No. DE-AC02-05CH11231.



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





**Table captions**
Table 1. List of simulations.





**Table 1.** List of simulations.

| Simulation | Years | Vertical Levels | Description |
|---|---|---|---|
| EAMv1 | 6 | 72 | Standard EAMv1 with the default deterministic ZM deep convection scheme for simulating the current climate[1] |
| STOCH | 6 | 72 | Same as EAMv1, but coupling with the PC stochastic deep convection scheme with the deterministic ZM deep convection scheme |
| EAMv1-30L | 6 | 30 | Same as EAMv1, but using a vertical resolution configuration of 30 layers |
| STOCH-30L | 6 | 30 | Same as STOCH, but using a vertical resolution configuration of 30 layers |
| EAMv1-4K | 3 | 72 | Same as EAMv1, but for simulating a warmer world[2] |
| STOCH-4K | 3 | 72 | Same as STOCH, but for simulating a warmer world |

[1]Atmosphere-only simulations, using fully prognostic atmosphere and land models with prescribed,
seasonally varying climatological present-day sea surface temperatures (SSTs) and sea ice extent,
recycled yearly.
[2]For simulating a warmer world, the atmosphere-only simulations are subjected to a composite
SST warming pattern derived from the Coupled Model Intercomparison Project Phase 3 (CMIP3)
coupled models.





**Figure captions**

**Figure 1.** Spatial distributions of the 20–80 day variance of rainfall from (a) the Xie-Arkin observations, (b) TRMM, (c) EAMv1, and (d) STOCH, respectively (units: $mm^2 d^{-2}$).

**Figure 2.** Spatial distributions of the synoptic variance of rainfall from (a) TRMM, (b) EAMv1, and (c) STOCH, respectively (units: $mm^2 d^{-2}$).

**Figure 3.** Frequency distributions of (a) total (solid line), (b) convective (solid line) and large-scale (dashed line) precipitation intensity over the tropics (20ºS, 20ºN) for EAMv1 (blue) and STOCH (red) respectively. For total precipitation, the TRMM observations (black) are included for evaluation.

**Figure 4.** Frequency distributions of total precipitation intensity over Amazon (20ºS-5ºN, 40ºW-80ºW), tropical western Pacific (TWP) (0ºN-15ºN, 130ºE-170ºE), India (14ºN-26.5ºN, 74.5ºE-94ºE; for June-September), Maritime Continent (MC) (10ºS-10ºN, 90ºE-160ºE), Southern Great Plains (SGP) (37ºN-42ºN, 90ºW-110ºW; for May-August) and eastern China (25ºN-35ºN, 100ºE-120ºE; for June-August) for TRMM (black), EAMv1 (blue) and STOCH (red) respectively.

**Figure 5.** Spatial distributions of frequencies of total rainfall intensity larger than (top row) 1 mm $d^{-1}$, (middle row) between 1 and 20 mm $d^{-1}$ and (bottom row) larger than 20 mm $d^{-1}$ for TRMM, EAMv1 and STOCH, respectively.

**Figure 6.** Annual mean rainfall amount distributions of (a) total precipitation (solid line) over the tropics (20ºS, 20ºN) for GPCP 1DD (grey), TRMM (black), EAMv1 (blue) and STOCH (red), respectively. Individual distributions of (b) convective (solid line) and large-scale (dashed line) precipitation in EAMv1 (blue) and STOCH (red) are also shown. The rainfall intensity on the x-axis is on a logarithmic scale with bin intervals of $\Delta \ln(R) = \Delta R/R = 0.1$.

**Figure 7.** Histogram of percentage frequency of total rainy events as a function of their duration using 3-hourly data (conditional probability of rainfall, given rainfall the previous times) from TRMM (black), EAMv1 (blue) and STOCH (red) for the threshold rainfall rate of 1 mm $d^{-1}$ over the tropics.

**Figure 8.** Same as Fig. 3, but including PDFs for EAMv1-30L and STOCH-30L (both dashed lines).

**Figure 9.** Joint PDFs of CAPE versus convective precipitation over the tropics (20ºS, 20ºN) from (a) EAMv1, (b) EAMv1-30L, (c) STOCH, and (d) STOCH-30L, respectively.

**Figure 10.** Frequencies of the resolved upward moisture flux over the tropics (20ºS, 20ºN) in EAMv1, EAMv1-30L, STOCH and STOCH-30L, respectively.





**Figure 11.** Global distributions of total precipitation for (a) GPCP, (b) EAMv1, and (c) STOCH,
and differences of (d) total, (e) convective and (f) large-scale precipitation between STOCH and
EAMv1. Differences with a confidence level greater than 95% in (d-f) are stippled.
**Figure 12.** Annual and zonal mean cross sections of (a-c) temperature and (d-f) specific humidity
for (a&d) ERAI and differences for (b&e) EAMv1-ERAI and (c&f) STOCH-EAMv1. Differences
with a confidence level greater than 95% in between STOCH and EAMv1 are stippled.
**Figure 13.** Taylor diagram with metrics for STOCH, compared with EAMv1.
**Figure 14.** Geographical distributions of responses of (a&b) annual mean precipitation and (c&d)
precipitation extremes (R95p) to climate warming from +4K experiments. Differences with a
confidence level greater than 95% are stippled.

## Figures

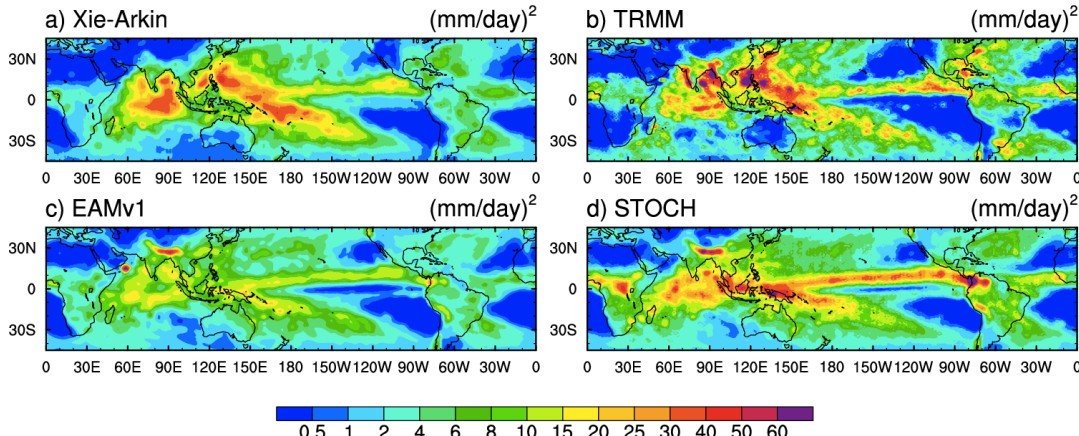

**Figure 1.** Spatial distributions of the 20–80 day variance of rainfall from (a) the Xie-Arkin observations, (b) TRMM, (c) EAMv1, and (d) STOCH, respectively (units: $mm^2\ d^{-2}$).



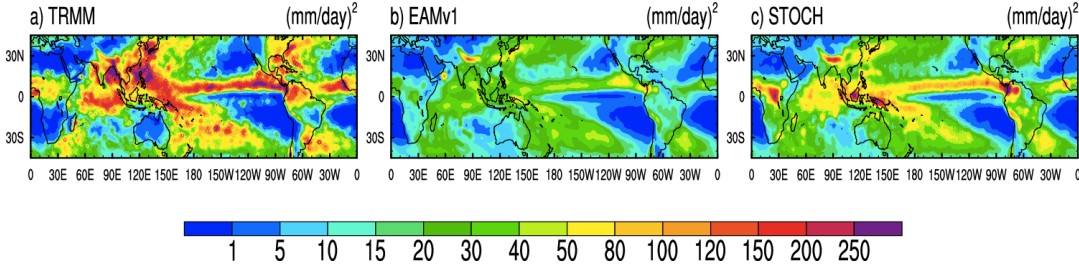

**Figure 2.** Spatial distributions of the synoptic variance of rainfall from (a) TRMM, (b) EAMv1,

and (c) STOCH, respectively (units: $mm^2\ d^{-2}$).





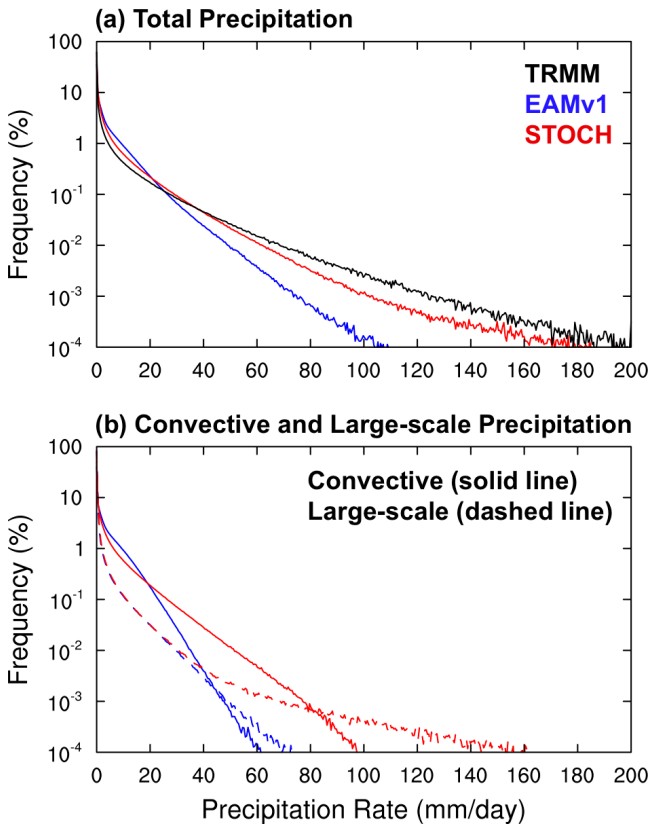

**Figure 3.** Frequency distributions of (a) total (solid line), (b) convective (solid line) and large-scale (dashed line) precipitation intensity over the tropics (20ºS, 20ºN) for EAMv1 (blue) and STOCH (red) respectively. For total precipitation, the TRMM observations (black) are included for evaluation.



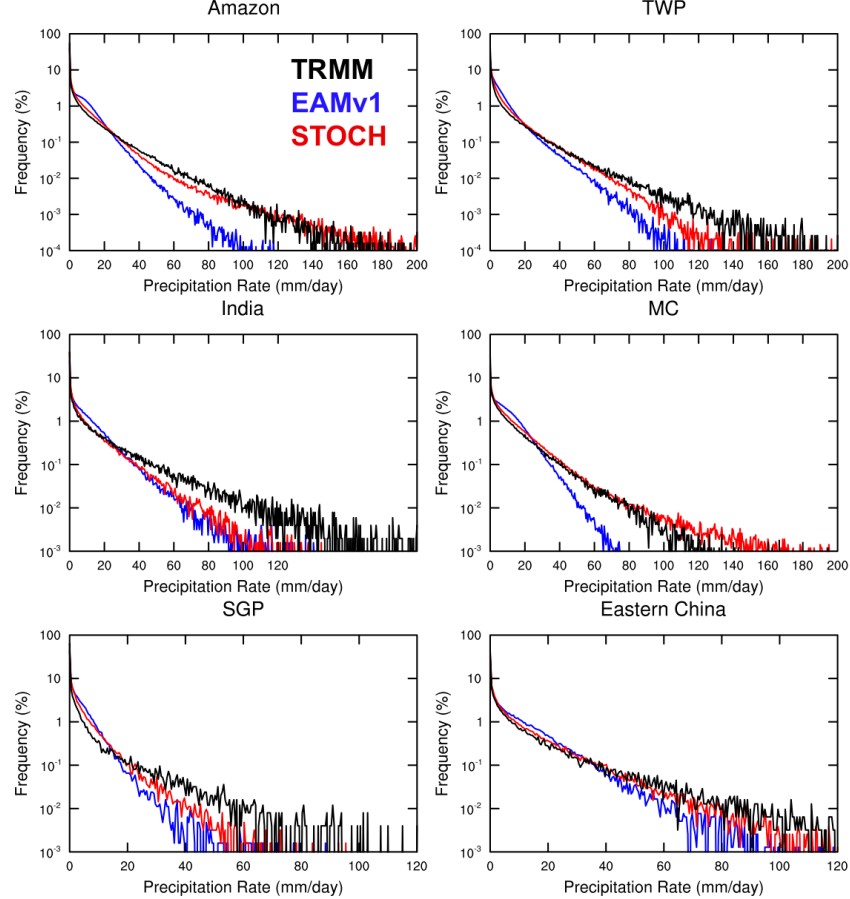

**Figure 4.** Frequency distributions of total precipitation intensity over Amazon (20ºS-5ºN, 40ºW-80ºW), tropical western Pacific (TWP) (0ºN-15ºN, 130ºE-170ºE), India (14ºN-26.5ºN, 74.5ºE-94ºE; for June-September), Maritime Continent (MC) (10ºS-10ºN, 90ºE-160ºE), Southern Great Plains (SGP) (37ºN-42ºN, 90ºW-110ºW; for May-August) and eastern China (25ºN-35ºN, 100ºE-120ºE; for June-August) for TRMM (black), EAMv1 (blue) and STOCH (red) respectively.



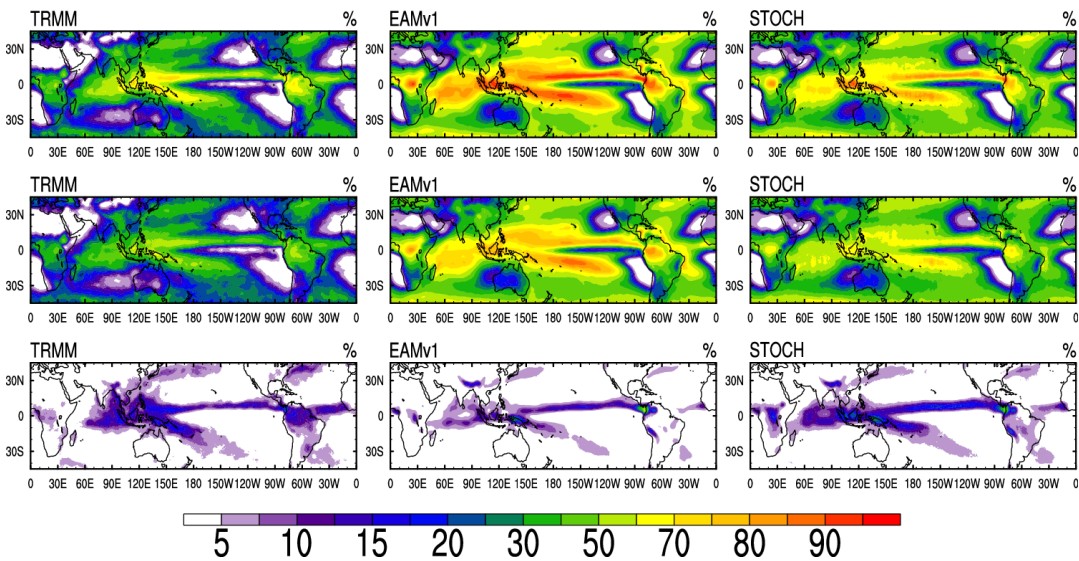

**Figure 5.** Spatial distributions of frequencies of total rainfall intensity larger than (top row) 1 mm d$^{-1}$, (middle row) between 1 and 20 mm d$^{-1}$ and (bottom row) larger than 20 mm d$^{-1}$ for TRMM, EAMv1 and STOCH, respectively.



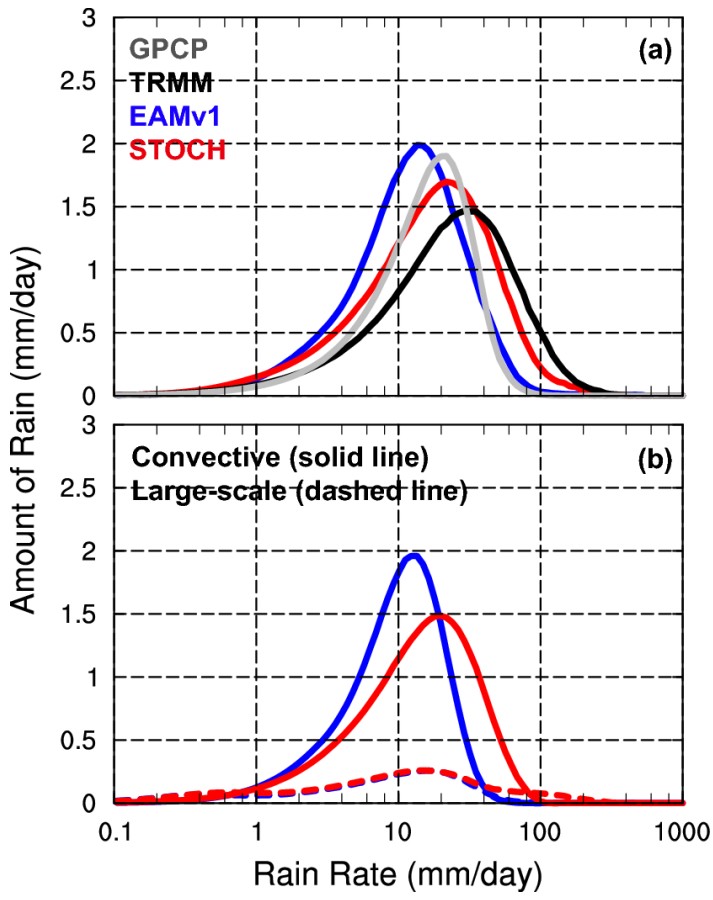

**Figure 6.** Annual mean rainfall amount distributions of (a) total precipitation (solid line) over the tropics (20ºS, 20ºN) for GPCP 1DD (grey), TRMM (black), EAMv1 (blue) and STOCH (red), respectively. Individual distributions of (b) convective (solid line) and large-scale (dashed line) precipitation in EAMv1 (blue) and STOCH (red) are also shown. The rainfall intensity on the x-axis is on a logarithmic scale with bin intervals of $\Delta \ln(R) = \Delta R/R = 0.1$.



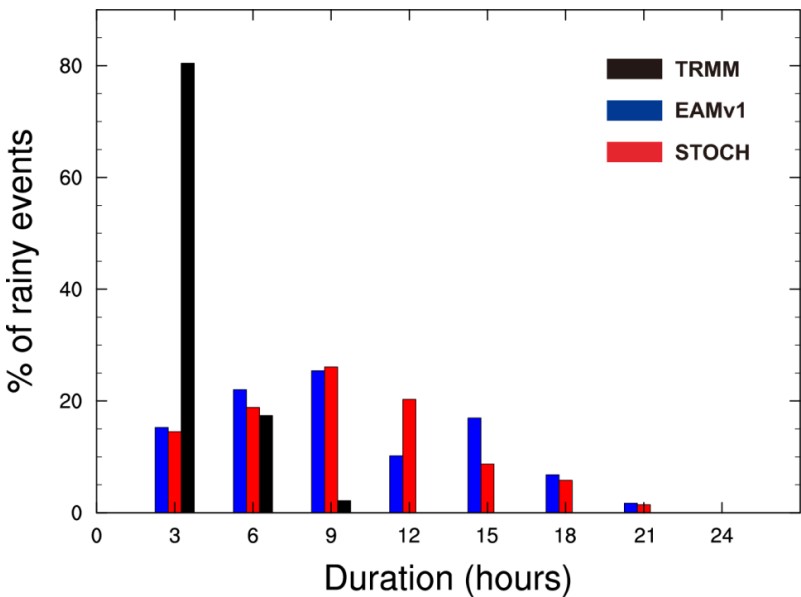

**Figure 7.** Histograms of percentage frequency of total rainy events as a function of their duration
using 3-hourly data (conditional probability of rainfall, given rainfall the previous times) from
TRMM (black), EAMv1 (blue) and STOCH (red) for the threshold rainfall rate of 1 mm d$^{-1}$ over
the tropics.



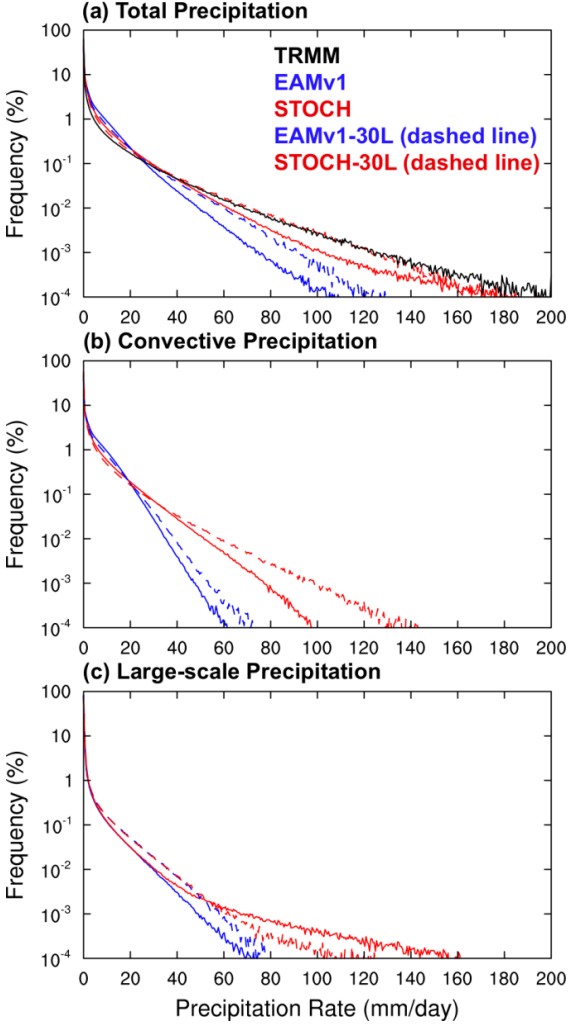

**Figure 8.** Same as Fig. 3, but including PDFs for EAMv1-30L and STOCH-30L (both dashed

lines).





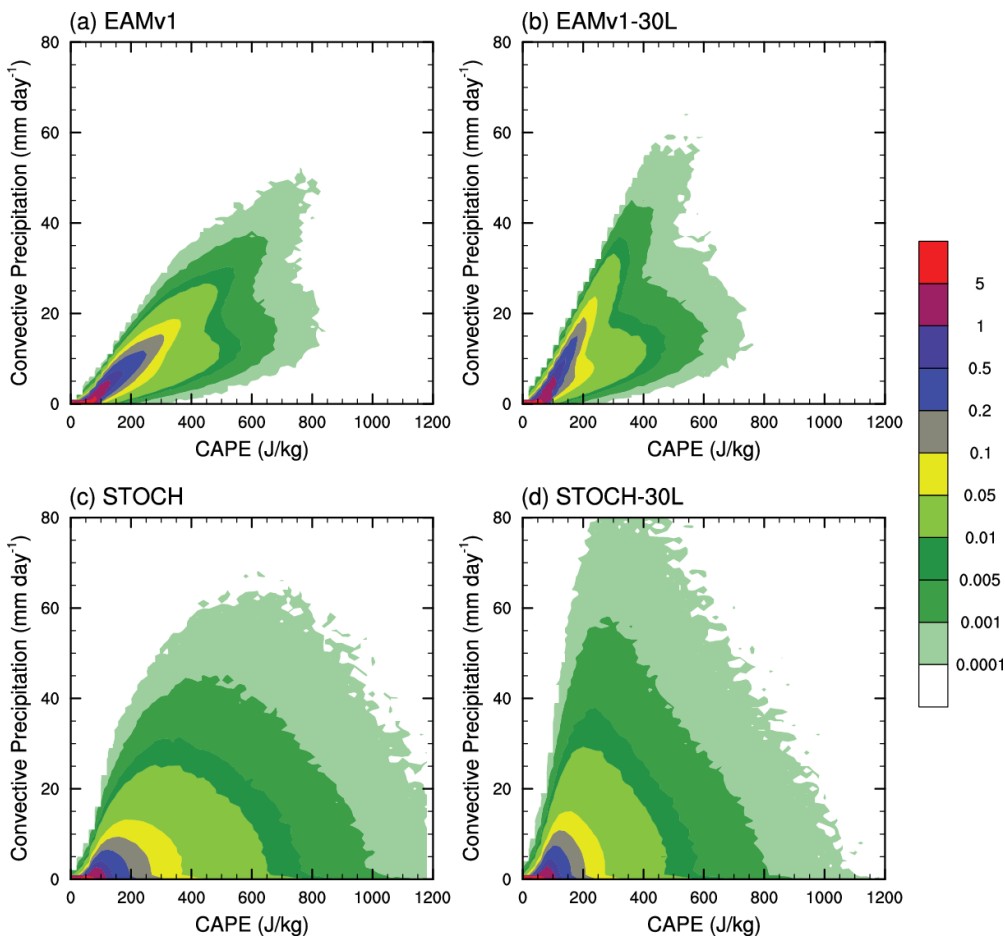

**Figure 9.** Joint PDFs of CAPE versus convective precipitation over the tropics (20ºS, 20ºN) from (a) EAMv1, (b) EAMv1-30L, (c) STOCH, and (d) STOCH-30L, respectively.



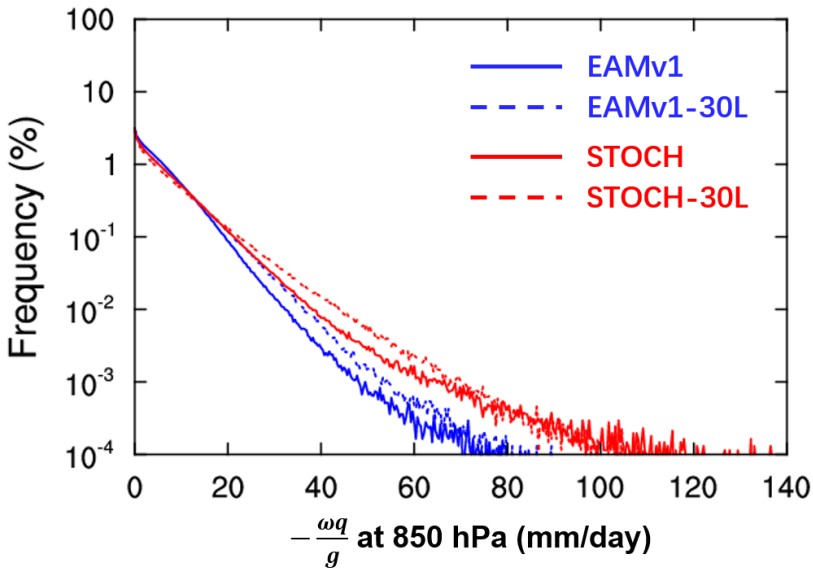

**Figure 10.** Frequencies of the resolved upward moisture flux over the tropics (20ºS, 20ºN) in EAMv1, EAMv1-30L, STOCH and STOCH-30L, respectively.



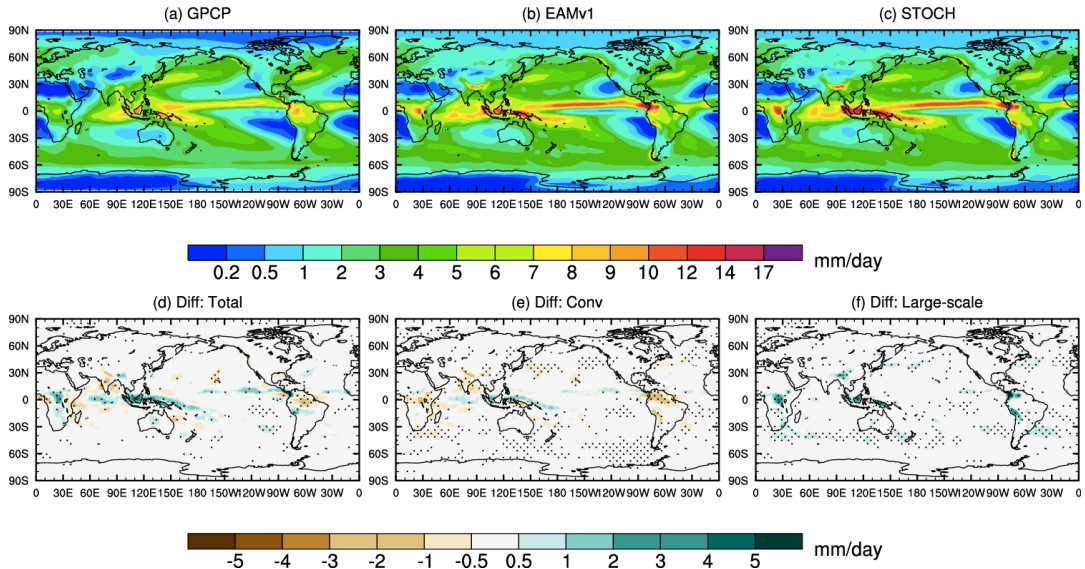

**Figure 11.** Global distributions of total precipitation for (a) GPCP, (b) EAMv1, and (c) STOCH, and differences of (d) total, (e) convective and (f) large-scale precipitation between STOCH and EAMv1. Differences with a confidence level greater than 95% in (d-f) are stippled.



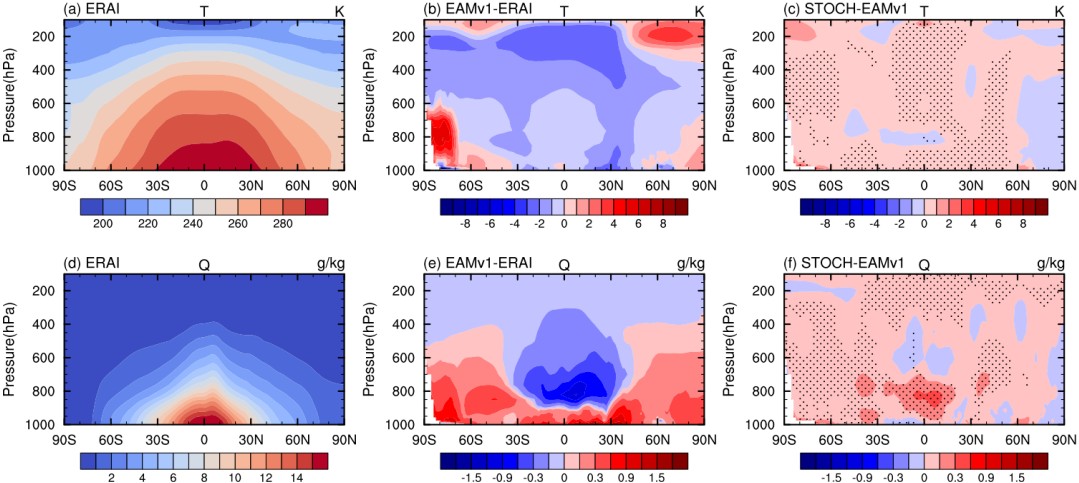

**Figure 12.** Annual and zonal mean cross sections of (a-c) temperature and (d-f) specific humidity for (a&d) ERAI and differences for (b&e) EAMv1-ERAI and (c&f) STOCH-EAMv1. Differences with a confidence level greater than 95% in between STOCH and EAMv1 are stippled.



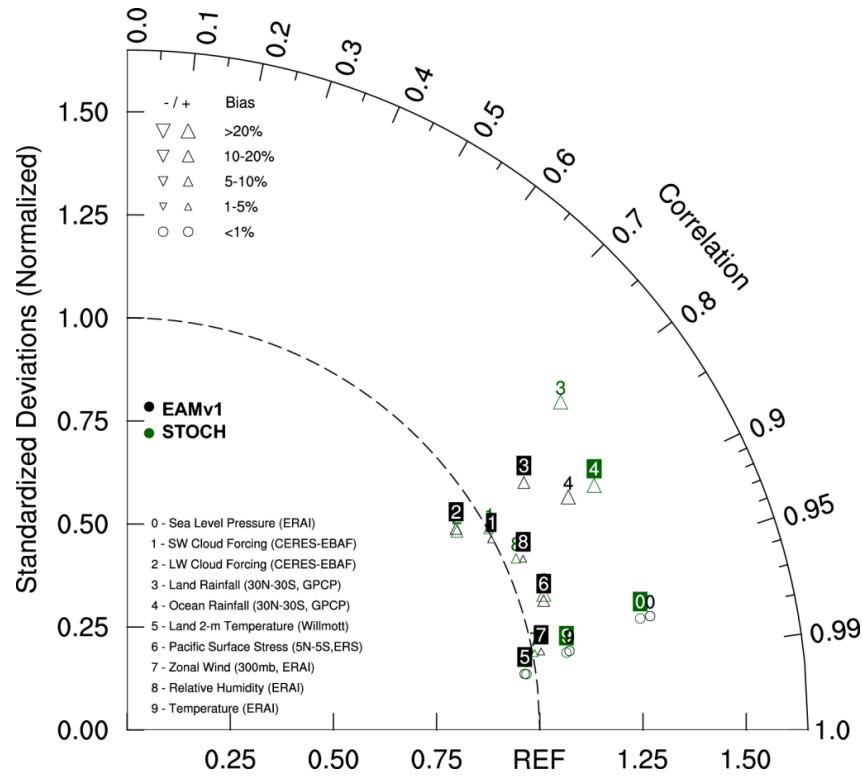

**Figure 13.** Taylor diagram with metrics for STOCH, compared with EAMv1.



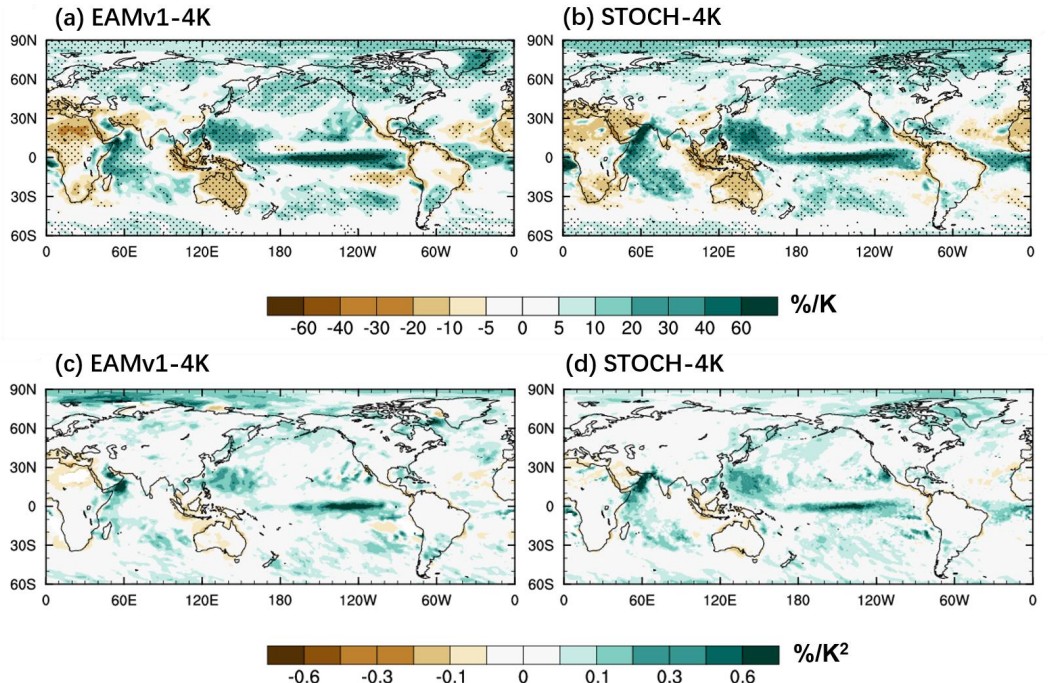

**Figure 14.** Geographical distributions of responses of (a&b) annual mean precipitation and (c&d)

precipitation extremes (R95p) to climate warming from +4K experiments. Differences with a

confidence level greater than 95% are stippled.