# Peer review of "Effects of Coupling a Stochastic Convective Parameterization"

_Geoscientific Model Development, 2020_

## Referee Comment (RC1) · Anonymous Referee #1 · 30 Oct 2020

**Reviewer's Summary of Manuscript**

In "Effects of Coupling a Stochastic Convective Parameterization with Zhang-McFarlane Scheme on Precipitation Simulation in the DOE E3SMv1 Atmosphere Model," Wang et al. document their modifications to the Plant-Craig 2008 (PC08) stochastic convective parameterization necessary for its incorporation in the Energy Exascale Earth System Model v1 (E3SM), and they document the performance of the model. The stochastic version of E3SM has higher frequency of extreme precipitation events and more precipitation contributed from extreme precipitations; these changes bring the model statistics closer to those observed in TRMM and GPCP. Large scale precipitation also becomes more intense, which the authors attribute to grid-scale CAPE being allow to accumulate between timesteps to due the randomness of when the stochastic parameterization does (or does not) trigger. The authors show that the use of the stochastic parameterization reduces the frequency of light rain rates (drizzle), since the stochastic parameterization tends to generate higher rain rates in general and because the intermittency reduces the frequency of convection overall.

The authors argue that the mean state of the model is essentially the same between the stochastic and control versions of E3SM. They suggest that further modifications to the parameterization would be necessary for use at higher resolution.

**Summary of Review**

Overall, the authors present a thorough overview of the stochastic E3SM's performance. They make insightful use of modeling experiments (e.g., running the default ZM parameterization along the stochastic version to determine why the stochastic version has less drizzle), and overall they provide a relatively strong case the that the use of the stochastic version improves the model performance.

The manuscript is generally well written, the chosen manuscript category of "development and technical papers" seems appropriate given the content, and the manuscript is clearly appropriate for GMD.

There are a number of additional analytic improvements that would substantially strengthen the manuscript: consideration of observational uncertainty in precipitation (especially for extremes), quantification of changes in the spatial structure of precipitation, inclusion of process-focused evaluation diagnostics, and (relatedly) analysis of the CAPE-precip relationship in observations.

In addition to these analytic issues, the manuscript needs to be modified in order to comply with the GMD code and data policy.

Given that the manuscript is clearly appropriate for GMD, and given that it would benefit from substantial additional analysis, I am recommending that the manuscript be returned to the authors for major revisions.

The major concerns noted above are described in more detail below, and the final section points out some specific, minor issues.

**Major Concerns**

**Precipitation observational uncertainty: FROGS**

The authors compare simulated precipitation with observations throughout their manuscript (i.e., Figures 1–8). Recent studies have shown that uncertainty in observed precipitation is quite large: especially in the tails of the distribution, which is where the authors focus most of their analysis. In most cases, the authors' analysis shows that the stochastic E3SM is closer to observations than the default; but would that picture be as clear if observational uncertainty were included?

Fortunately, a recent effort (published in a related EGU journal in fact) has produced a dataset of essentially all existing daily precipitation datasets on a common 1x1-degree grid: see Roca et al. (2019). Therefore it should be relatively simple for the authors to investigate and/or show the impact of observational uncertainty on their main conclusions.

Roca, R., Alexander, L. V., Potter, G., Bador, M., Juca, R., Contractor, S., Bosilovich, M. G., and Cloche, S.: FROGS: a daily 1×1 gridded precipitation database of rain gauge, satellite and reanalysis products, Earth Syst. Sci. Data, 11, 1017–1035, https://doi.org/10.5194/essd-11-1017-2019, 2019.

**Process-focused diagnostics**

The authors do a good job of assessing the statistics of precipitation and convection in stochastic E3SM, but I was a bit surprised that the authors never showed any process-focused comparisons of the model's convection with observations. Given that this is a DOE-funded effort, I could imagine–for example–performing hindcast simulations and comparing with the extensive atmospheric observations provided by ARM: does the hindcast CAPE improve, does the hindcast precipitation improve, etc. I suspect that this shouldn't be too difficult, since another DOE-funded author has provided a framework for generating hindcasts with E3SM (see https://github.com/zarzycki/betacast).

Alternatively or additionally, it might be useful to look at the convective pickup relationship, which has increasingly been used as a bulk diagnostic of tropical convection in models: e.g., see Kuo et al. (2018). At the very least, I would imagine that the stochastic parameterization increases spread in the precipitable water-precipitation relationship, but I wonder if it improves the model relative to observations or not?

Kuo, Y. H., K. A. Schiro, and J. D. Neelin, 2018: Convective transition statistics over tropical oceans for climate model diagnostics: Observational baseline. J.

Atmos. Sci., 75, 1553–1570, https://doi.org/10.1175/JAS-D-17-0287.1.

**Quantification of spatial structure of variability**

Related to the above, in playing around with the model output (see images below), I noticed that the spatial patterns of precipitation were substantially (and unsurprisingly) noiser in the stochastic version. It struck me that the texture of the precipitation field seemed somewhat unrealistic. It would be useful to compare the simulated precipitation with observed precipitation: especially if the authors are able to run a hindcast simulation.

[Figure]

[Figure]

Alternatively, it might be useful to quantitatively examine the spatial statistics of precipitation: e.g., looking at structure functions (aka variograms), or equivalently power spectra. My concern is that the stochastic parameterization introduces too much small-scale noise, and if so, this should be quantified and documented.

**CAPE-precip relationship in observations**

Related to the above, in Figure 9 the authors use the CAPE-precip relationship to argue why the statistics change with vertical resolution. This seemed like an ideal opportunity to compare this process-oriented relationship with observations: is the dramatic change in the CAPE-precip relationship an improvement relative? Was the bimodality in the CAPE-precip relationship in the control simulation a realistic feature (presumably not)?

This should be relatively easy for the authors to reproduce, given that multiple reanalysis datasets reside in a semi-public location on the DOE-funded NERSC CFS filesystem in association with the CMIP6 data that have been collected for the DOE community. If the authors are unfamiliar with how to access this repository at NERSC, their program manager can likely help direct them.

**Code and data availability and appropriate use of archival repositories**

The authors do point to the E3SM code in their "Code and data availability" section, but there are two fundamental issues with the way that the authors have done this. First, it does not appear that the code for the stochastic version of the ZM parameterization is available in the public release of E3SM, or if it is, it is not in an obvious location. The GMD policy (see (https://www.geoscientific-model-development.net/about/manuscript_types.html#item2)[https://www.geoscientific-model-development.net/about/manuscript_types.html#item2]) requies that "In the case where new code is described in the paper... The code should be made available."

The code and data policy requires that code be archived in a long-term repository, and it explicitly states that github is not appropriate for this purpose. Similar to what the authors have done with the EAMv1 data, they should upload the E3SM code–that was used for this study–and they should amend the "Code and data availability" to make it clear where the new code resides in the repository.

**Metadata in deposited code**

In checking out the EAMv1 data that the authors uploaded to Zenodo, I noticed that the data files lack the lat/lon arrays; this effectively renders the data files unusable unless a person knows somewhere to find the lat/lon data for this model configuration. The authors should upload new versions of the files with the lat/lon fields added.

**Specific, minor issues**

**Model version**

I think that the title of the paper should also be amended to reflect the specific version of E3SM that was used. In browsing through the E3SM tags, I see that several iterations of the E3SMv1 model exist (e.g., v1.1.0), so the use of "E3SMv1" is ambiguous.

**Misinterpretation of O'Brien et al., (2016)**

On lines 359–362, the authors state that the vertical resolution dependence they see is consistent with that shown in O'Brien et al., (2016). I believe this is a misinterpretation of O'Brien et al. (2016). Their resolution-dependence result is described in more detail by Rauscher et al. (2016); in Equation (2) of Rauscher et al., there is a term related to the vertical grid spacing. Once terms are rearranged to solve for $W$, this yields the relationship $W \propto \Delta p$. Therefore, the result of Rauscher et al., (2016) and O'Brien et al., (2016) implies that vertical velocity should *increase* as vertical grid spacing increases. Figure 10 seems to

show higher moisture flux values for the 30L simulations, which is consistent with this theory. However, this appears to be inconsistent with the language used in the manuscript, which tangentially states that "refining horizontal resolution should result in more large-scale precipitation."

Rauscher, S. A., T. A. O'Brien, C. Piani, E. Coppola, F. Giorgi, W. D. Collins, and P. M. Lawston, 2016: A multimodel intercomparison of resolution effects on precipitation: simulations and theory. Clim. Dyn., 47, 2205–2218, https://doi.org/10.1007/s00382-015-2959-5.

**Grammar and syntax**

There are a number of few places in the manuscript with grammaer and syntax issues. I recommend that the manuscript be thoroughly proof-read by someone outside the authorship team prior to resubmitting.

---

## Referee Comment (RC2) · Anonymous Referee #2 · 6 Nov 2020

**Review of "Effects of Coupling a Stochastic Convective Parameterization with Zhang-McFarlane Scheme on Precipitation Simulation in the DOE E3SMv1 Atmosphere Model" by Wang et al.**

This paper by Wang et al. demonstrates the effects of incorporating a stochastic convective scheme, primarily on model precipitation statistics. The Zhang-McFarlane (ZM) convective scheme in the DOE E3SM is modified such that the subgrid-scale convection responds stochastically to grid-scale forcings. Simulations with the stochastic scheme are compared to simulations with a deterministic ZM scheme. The stochastic scheme compares more favorably to observations in several measures of tropical precipitation distribution. Most notably, the frequency of light rain is reduced while the frequency of heavy rain is increased. This increased precipitation variability does not appear at the expense of model mean state degradations. The behavior of the mean and extreme precipitation under warming is nearly unchanged with the stochastic scheme. The authors also highlight a resolution dependence of the stochastic scheme. A lower vertical resolution model displays greater increases in the tails of precipitation distribution when compared to higher resolution models.

Overall, the paper is well-written, and contains results that are interesting and worthy of dissemination. I do have a short stack of issues that must be addressed prior to publication. The introductory text on the technical details of the convective scheme is too opaque. One or two figures and related arguments appear suspect, and need some thinking through. Please see specific comments below.

**Major comments**

1. **Section 2.1**. For someone with no background knowledge about the Plant and Craig scheme (like me), the text in this subsection is inadequate to explain how it works. I found myself repeatedly referring back to PC08 and Wang et al. 2016. I understand that explanations have appeared in these predecessor papers, but a little organization or perhaps a schematic would help the reader.

   (a) In **eq. 1, Line 110**: it would be helpful to point out at the outset, which of the variables $\langle m \rangle$, $\langle M \rangle$ and $\langle N \rangle$ are coming from the deterministic portion of the ZM scheme. One has to read until **Line 132** or refer to other papers to know that $\langle m \rangle$ is fixed.

   (b) The integral over the 'probability' in eq. 1 does not equal 1, i.e.,

   $$\frac{\langle N \rangle}{\langle m \rangle} \int_0^\infty e^{-\frac{m}{\langle m \rangle}} dm = \langle N \rangle.$$

   Presumably $\langle N \rangle > 1$, so this is not a true probability, but actually denotes the mean number of clouds with mass flux between $m$ and $m + dm$. However, this measure is being compared against a random number between 0 and 1 to generate

the subgrid-scale spectrum of clouds. If this is indeed a true probability, then the right normalization is perhaps $\langle M \rangle$. At this point it is unclear if this is a typographical error or a case of inadequate information or a structural error in the implementation of the convective scheme. Please resolve this apparent inconsistency.

(c) **Line 122**: 'quasi-equilibrium' suddenly appears with no context. This is an important concept that deserves a little more attention. Move this closer to eq. 1 and at least mention that the quasi-equilibrium assumption yields the mass-flux $\langle M \rangle$ from the closure in the ZM scheme.

(d) Please include the details of how the convective tendencies are produced. In PC08, a plume model is used to compute the temperature and moisture tendencies for random number of clouds with a different cloud base mass-fluxes. These tendencies are then combined to generate ensemble-mean tendencies. Is a similar plume model used here? Is it different from the plume model within the ZM scheme, which—somewhat confusingly—generates its own spectrum of entraining plumes?

2. The parameter $\langle m \rangle$ in the PC08 scheme appears to control the subgrid-scale mass-flux distribution, but is held fixed in this study. It would be useful if the authors could state whether and how the results are sensitive to perturbations in $\langle m \rangle$ (or any other parameter that controls the subgrid-scale variance). One would expects the rain-rate pdfs, particularly the tails of the distribution, to be impacted. A large sensitivity should be quantified and documented in this paper, a small sensitivity can simply be mentioned in text.

3. Figure 10 and interpretation: The reasoning in this analysis is not clear. The fact that precipitation is related to the low-level vertical moisture transport is not surprising because those two terms nearly equal each other in heavily raining situations. Can one really jump to a causality argument and claim that $-\omega q/g$ 'explains' the large-scale precipitation pdfs in Fig. 8? Moreover, if one insists on using $-\omega q/g$, it remains to explained why the low-level q flux pdfs in Fig. 10 do not exactly correspond to the large-scale precipitation pdfs in Fig. 8. For instance, in Fig. 10, the tails ($>70$ mm/d) of the STOCH and STOCH-30L q flux pdfs are nearly coincident, which is quite different from the divergent tails of the large-scale precipitation PDFs in Fig. 8. I suggest that the authors remove this figure and interpretation, as it muddies more than clarifies.

**Minor comments**

1. Line 365: Mention if this the dilute CAPE or the non-entraining CAPE.

2. Line 381: Remove the 'an' in 'no longer an approximately linear relations'

3. Figures 1 and 2: there is excessive precipitation variance over a few land regions: central Africa, the Himalayas, the Maritime Continent and near the Colombian coast. If not explained, this should at least be be pointed out.

4. Figure 9 and interpretation. This figure does not explain why the convective precipitation pdfs are different between EAMv1 and EAMv1-30L. All we know is that EAMv1-30L generates tends to generate higher precipitation values for a given CAPE values (but why?). However, it is nice to see how the stochasticity affects the CAPE-precipitation relationship, so I suggest that the authors keep this figure, but change the interpretation.

5. Figures 14c and d: There are some quantitative differences between STOCH and EAMv1 that are harder to gauge with the coefficient $a$ from equation 8. I suggest that the authors also show the fractional change in $r_x$ such that the units are in %/K. In addition, it would also be useful to show the tropical precipitation pdfs like in Figure 3 to verify that the pdfs get stretched by nearly the same factor with and without the stochastic parameterization.

---

## Author Comment (AC1) · 28 Jan 2021

**Response to Reviewers Comments**

**Reply to the comments by Referee #2**

We thank the reviewer for the valuable comments and suggestions on improving the manuscript. Below is our point-by-point response to these comments. The reviewer's comments are in italic and our responses are in normal font.

*This paper by Wang et al. demonstrates the effects of incorporating a stochastic convective scheme, primarily on model precipitation statistics. The Zhang-McFarlane (ZM) convective scheme in the DOE E3SM is modified such that the subgrid-scale convection responds stochastically to grid-scale forcings. Simulations with the stochastic scheme are compared to simulations with a deterministic ZM scheme. The stochastic scheme compares more favorably to observations in several measures of tropical precipitation distribution. Most notably, the frequency of light rain is reduced while the frequency of heavy rain is increased. This increased precipitation variability does not appear at the expense of model mean state degradations. The behavior of the mean and extreme precipitation under warming is nearly unchanged with the stochastic scheme. The authors also highlight a resolution dependence of the stochastic scheme. A lower vertical resolution model displays greater increases in the tails of precipitation distribution when compared to higher resolution models.*

*Overall, the paper is well-written, and contains results that are interesting and worthy of dissemination. I do have a short stack of issues that must be addressed prior to publication. The introductory text on the technical details of the convective scheme is too opaque. One or two figures and related arguments appear suspect, and need some thinking through. Please see specific comments below.*
**Reply:** We thank the reviewer for the positive remarks.

*Major Concerns*

*Section 2.1. For someone with no background knowledge about the Plant and Craig scheme (like me), the text in this subsection is inadequate to explain how it works. I found myself repeatedly referring back to PC08 and Wang et al. 2016. I understand that explanations have appeared in these predecessor papers, but a little organization or perhaps a schematic would help the reader.*

*(a) In eq. 1, Line 110: it would be helpful to point out at the outset, which of the variables $\langle m \rangle$, $\langle M \rangle$ and $\langle N \rangle$ are coming from the deterministic portion of the ZM scheme. One has to read until Line 132 or refer to other papers to know that $\langle m \rangle$ is fixed.*

**Reply:** We will revise the manuscript to make this clearer by expanding the description of the stochastic convection. For the reviewer's information here, **<m>** is the mean mass flux of a cloud and is a prescribed parameter, set to $1 \times 10^7$ kg s$^{-1}$. **<M>** as referred to in line 114 in the original manuscript is the ensemble cloud mass flux, obtained from the closure of the deterministic ZM scheme. **<N>** is the mean number of clouds, equal to **<M>/<m>**.

*(b) The integral over the 'probability' in eq. 1 does not equal 1, i.e.,*

$$\frac{\langle N \rangle}{\langle m \rangle} \int_0^\infty e^{-\frac{m}{\langle m \rangle}} dm = \langle N \rangle.$$

*Presumably $\langle N \rangle > 1$, so this is not a true probability, but actually denotes the mean number of clouds with mass flux between m and m + dm. However, this measure is being compared against a random number between 0 and 1 to generate the subgrid-scale spectrum of clouds. If this is indeed a true probability, then the right normalization is perhaps $\langle M \rangle$. At this point it is unclear if this is a typographical error or a case of inadequate information or a structural error in the implementation of the convective scheme. Please resolve this apparent inconsistency.*

**Reply:** We thank the reviewer for pointing this out. The implementation is correct. It is just that we did not describe it accurately in the text. The probability we compare the random number with is

$$\frac{1}{<m>} e^{-\frac{m}{<m>}} dm$$

Then the sum of mass fluxes generated this way is multiplied by the factor *<N>* to rescale it to the mass flux of all clouds.

*(c) Line 122: 'quasi-equilibrium' suddenly appears with no context. This is an important concept that deserves a little more attention. Move this closer to eq. 1 and at least mention that the quasi-equilibrium assumption yields the mass-flux $\langle M \rangle$ from the closure in the ZM scheme.*

**Reply:** Done.

*(d) Please include the details of how the convective tendencies are produced. In PC08, a plume model is used to compute the temperature and moisture tendencies for random number of clouds with a different cloud base mass-fluxes. These tendencies are then combined to generate ensemble-mean tendencies. Is a similar plume model used here? Is it different from the plume model within the ZM scheme, which—somewhat confusingly—generates its own spectrum of entraining plumes?*

**Reply:** Yes, a similar plume model is used here; it is the bulk plume model of the ZM scheme, which gives the vertical profiles of heating and drying per unit cloud base mass flux. After summing up cloud base mass fluxes from all convective clouds successfully launched by comparing individual probabilities with random numbers, we obtain the total cloud base mass flux.

The product of this total mass flux and the temperature and moisture tendencies form the bulk plume model gives the final temperature and moisture tendencies by the subgrid convective clouds.

*Figure 10 and interpretation: The reasoning in this analysis is not clear. The fact that precipitation is related to the low-level vertical moisture transport is not surprising because those two terms nearly equal each other in heavily raining situations. Can one really jump to a causality argument and claim that −ωq/g 'explains' the large- scale precipitation pdfs in Fig. 8? Moreover, if one insists on using −ωq/g, it remains to explained why the low-level q flux pdfs in Fig. 10 do not exactly correspond to the large-scale precipitation pdfs in Fig. 8. For instance, in Fig. 10, the tails (>70 mm/d) of the STOCH and STOCH-30L q flux pdfs are nearly coincident, which is quite different from the divergent tails of the large-scale precipitation PDFs in Fig. 8. I suggest that the authors remove this figure and interpretation, as it muddies more than clarifies.*

**Reply:** Following the reviewer's suggestion, Figure 10 and related interpretation have been removed in the revised manuscript.

*Minor comments*
1.  *Line 365: Mention if this the dilute CAPE or the non-entraining CAPE.*

**Reply:** Dilute CAPE

2.  *Line 381: Remove the 'an' in 'no longer an approximately linear relations'*

**Reply:** Done.

*Figures 1 and 2: there is excessive precipitation variance over a few land regions: central Africa, the Himalayas, the Maritime Continent and near the Colombian coast. If not explained, this should at least be pointed out.*

**Reply:** We will point out these degradations in the revision.

*Figure 9 and interpretation. This figure does not explain why the convective precipitation pdfs are different between EAMv1 and EAMv1-30L. All we know is that EAMv1-30L generates tends to generate higher precipitation values for a given CAPE values (but why?). However, it is nice to see how the stochasticity affects the CAPE- precipitation relationship, so I suggest that the authors keep this figure, but change the interpretation.*

**Reply:** A coarser vertical resolution means stronger vertical mixing resulting in higher precipitation. The interpretation is added in the revision.

*Figures 14c and d: There are some quantitative differences between STOCH and EAMv1 that are harder to gauge with the coefficient a from equation 8. I suggest that the authors also show the fractional change in $r_x$ such that the units are in %/K. In addition, it would also be useful to show the tropical precipitation pdfs like in Figure 3 to verify that the pdfs get stretched by nearly the same factor with and without the stochastic parameterization.*

**Reply:** Following the reviewer's suggestions, we show the fractional change in $r_x$ normalized by the global-mean surface air warming in Fig. R1. Consistent with Fig. 14, the spatial pattern and magnitude in the two simulations resemble each other. It also tells that the resemblance of the coefficient $a$ from Eq. (8) between the two simulations as shown in Fig. 14c&d results from the similar response of the fractional change in $r_x$ to global warming. Taking the tropical western Pacific (5ºN-20ºN; 130ºE-170ºE) where the maximum fractional increase in $r_x$ emerges for example, as climate warms, the pdfs in EAMv1_4k and STOCH_4k also get stretched by about the same amount. Fig. R1 has been merged into Fig. 14 as Fig. 14e and f in the revision.

[Figure]

**Fig. R1.** Geographical distributions of the fractional change in $r_x$ normalized by the global-mean surface air warming from +4K experiments.

[Figure]

**Fig. R2.** Frequency distributions of total precipitation intensity over the tropical western Pacific (5ºN-20ºN; 130ºE-170ºE) for EAMv1, STOCH, EAMv1-4K, and STOCH-4K.

---

## Author Comment (AC2) · 28 Jan 2021

**Response to Reviewers Comments**

**Reply to the comments by Referee #1**

We thank the reviewer for the constructive comments and suggestions on improving the manuscript. Below is our point-by-point response to these comments. The reviewer's comments are in italic and our responses are in normal font.

*Reviewer's Summary of Manuscript*

*In "Effects of Coupling a Stochastic Convective Parameterization with Zhang- McFarlane Scheme on Precipitation Simulation in the DOE E3SMv1 Atmosphere Model," Wang et al. document their modifications to the Plant-Craig 2008 (PC08) stochastic convective parameterization necessary for its incorporation in the Energy Exascale Earth System Model v1 (E3SM), and they document the performance of the model. The stochastic version of E3SM has higher frequency of extreme precipitation events and more precipitation contributed from extreme precipitations; these changes bring the model statistics closer to those observed in TRMM and GPCP. Large scale precipitation also becomes more intense, which the authors attribute to grid-scale CAPE being allow to accumulate between timesteps to due the randomness of when the stochastic parameterization does (or does not) trigger. The authors show that the use of the stochastic parameterization reduces the frequency of light rain rates (drizzle), since the stochastic parameterization tends to generate higher rain rates in general and because the intermittency reduces the frequency of convection overall.*
*The authors argue that the mean state of the model is essentially the same between the stochastic and control versions of E3SM. They suggest that further modifications to the parameterization would be necessary for use at higher resolution.*

*Summary of Review*

*Overall, the authors present a thorough overview of the stochastic E3SM's performance. They make insightful use of modeling experiments (e.g., running the default ZM parameterization along the stochastic version to determine why the stochastic version has less drizzle), and overall they provide a relatively strong case the that the use of the stochastic version improves the model performance.*

*The manuscript is generally well written, the chosen manuscript category of "development and technical papers" seems appropriate given the content, and the manuscript is clearly appropriate for GMD.*

*There are a number of additional analytic improvements that would substantially strengthen the manuscript: consideration of observational uncertainty in precipitation (especially for extremes), quantification of changes in the spatial structure of precipitation, inclusion of process-focused evaluation diagnostics, and (relatedly) analysis of the CAPE-precip relationship in observations. In addition to these analytic issues, the manuscript needs to be modified in order to comply with the GMD code and data policy.*

*Given that the manuscript is clearly appropriate for GMD, and given that it would benefit from substantial additional analysis, I am recommending that the manuscript be returned to the authors for major revisions.*

*The major concerns noted above are described in more detail below, and the final section points out some specific, minor issues.*

*Major Concerns*

*Precipitation observational uncertainty: FROGS*

*The authors compare simulated precipitation with observations throughout their manuscript (i.e., Figures 1–8). Recent studies have shown that uncertainty in observed precipitation is quite large: especially in the tails of the distribution, which is where the authors focus most of their analysis. In most cases, the authors' analysis shows that the stochastic E3SM is closer to observations than the default; but would that picture be as clear if observational uncertainty were included?*

*Fortunately, a recent effort (published in a related EGU journal in fact) has produced a dataset of essentially all existing daily precipitation datasets on a common 1x1-degree grid: see Roca et al. (2019). Therefore it should be relatively simple for the authors to investigate and/or show the impact of observational uncertainty on their main conclusions.*

*Roca, R., Alexander, L. V., Potter, G., Bador, M., Juca, R., Contractor, S., Bosilovich, M. G., and Cloche, S.: FROGS: a daily 1×1 gridded precipitation database of rain gauge, satellite and reanalysis products, Earth Syst. Sci. Data, 11, 1017–1035, https://doi.org/10.5194/essd-11-1017-2019, 2019.*

**Reply:** Thanks for your suggestion and for bringing our attention to this paper. The following rainfall products (summarized in Table. R1) are used to estimate the precipitation uncertainty in observations. They are almost the same as those used in Roca (2019) except the MSWEP version 2.2 which is currently not available from FROGAS, GPCC Full Daily v2018, GPCC First Guess v1 and CHIRPS v2.0 all for land only. Instead, GPM is used. As you mentioned, the uncertainty is larger for intense precipitation than for light precipitation (Fig. R1), which is also consistent with that in Roca (2019). Despite the uncertainties in observations, the simulated frequencies in STOCH are more consistent with those in the ensemble mean of all observations than those in the default EAMv1. Especially for light rain, the frequencies in STOCH fall in the observational range while those in EAMv1 do not. The product whose frequency is even lower than those in EAMv1 is GPCP; it is known to have underestimated precipitation intensities (Kooperman et al., 2016). Figure 3a has been replaced with Fig. R1 in the revision.

**Table R1.** List of gridded products and their acronyms

| Product shortname | Product name and version | Use of rain gauges data | Use of IR satellite data | Use of MW satellite data | References |
|---|---|---|---|---|---|
| TAPR | TAPEER v1.5 | No | Yes | multiple platforms | Roca et al. (2018) |

| TMPA | 3B42 v7.0 | Yes | Yes | multiple platforms | Huffman et al. (2009) |
|---|---|---|---|---|---|
| GSMArtg | GSMAP-NRT-gauges v6.0 | Yes | Yes | multiple platforms | Kubota et al. (2007) |
| PERS | PERSIANN CDR v1 r1 | Yes | Yes | No | Ashouri et al. (2015) |
| CMORg | CMORPH V1.0, CRT | Yes | Yes | multiple platforms | Xie et al. (2017) |
| GPCP | GPCP 1DD CDR v1.3 | Yes | Yes | One platform | Huffman et al. (2001) |
| GPM | GPM IMERG V06B | Yes | Yes | multiple platforms | Huffman et al. (2015) |

[Figure]

**Figure R1.** Frequency distributions of total precipitation intensity over the tropics (20ºS, 20ºN) for EAMv1 (blue), STOCH (red), TRMM (black) and ensemble mean of the observations listed in Table R1 (Obs_ens, purple) where each observation is denoted by the gray line.

References:

Roca, R.: Estimation of extreme daily precipitation thermodynamic scaling using gridded satellite precipitation products over tropical land. *Environmental Research Letters* **14**, 095009, doi:10.1088/1748-9326/ab35c6 (2019).

Kooperman, G. J., Pritchard, M. S., Burt, M. A., Branson, M. D., and Randall, D. A.: Robust effects of cloud superparameterization on simulated daily rainfall intensity statistics across multiple versions of the Community Earth System Model, J*ournal of Advances in Modeling Earth Systems* **8**, 140-165, (2016).

*Process-focused diagnostics*

*The authors do a good job of assessing the statistics of precipitation and convection in stochastic E3SM, but I was a bit surprised that the authors never showed any process-focused comparisons of the model's convection with observations. Given that this is a DOE-funded effort, I could imagine–for example–performing hindcast simulations and comparing with the extensive atmospheric observations provided by ARM: does the hindcast CAPE improve, does the hindcast precipitation improve, etc. I suspect that this shouldn't be too difficult, since another DOE-funded author has provided a framework for generating hindcasts with E3SM (see https://github.com/zarzycki/betacast).*

*Alternatively or additionally, it might be useful to look at the convective pickup relationship, which has increasingly been used as a bulk diagnostic of tropical convection in models: e.g., see Kuo et al. (2018). At the very least, I would imagine that the stochastic parameterization increases spread in the precipitable water-precipitation relationship, but I wonder if it improves the model relative to observations or not?*

*Kuo, Y. H., K. A. Schiro, and J. D. Neelin, 2018: Convective transition statistics over tropical oceans for climate model diagnostics: Observational baseline. J. Atmos. Sci., 75, 1553–1570, https://doi.org/10.1175/JAS-D-17-0287.1.*

**Reply:** We agree with the reviewer that hindcasts can provide process-focused diagnostics along with ARM observations. However, it is a significant amount of effort. Since this paper aims to provide the overall evaluation of the performance of the stochastic deep convection scheme on precipitation simulation as well as other climate mean states at the global scale, hindcast simulations are beyond the scope of this study. It will be done in a future paper. Instead, following the reviewer's other suggestion, we examined the precipitation pickup in the simulated precipitable water-precipitation relationship, as shown in Fig. R2, following Kuo et al. Over the tropical western Pacific (WPac), precipitation for the simulated column-integrated saturation humidity from 52 to 83.5 mm initiates at column water vapor (CWV) between 20 and 40 mm in EAMv1 which is smaller than that in observations (Kuo et al., 2018), implying much light rain is from small CWV. In comparison, the pickup of precipitation in STOCH is shifted to >40 mm, closer to observations. In the other three regions, the overall behavior of precipitation initiation between EAMv1 and STOCH for small and moderate column-integrated saturation humidity is comparable. Since more intense precipitation is simulated in STOCH than that in EAMv1, the precipitation rate at large CWV in STOCH is larger than that in EAMv1 agreeing better with the observations in Kuo et al. (2018).

[Figure]

**Figure R2.** Conditionally averaged precipitation rate as a function of column water vapor (CWV) and column-integrated saturation humidity (represented by colored dots) over the tropical (20ºS, 20ºN) western Pacific (WPac), the tropical eastern Pacific (EPac), Atlantic and Indian Ocean for (left) EAMv1 and (right) STOCH.

**Quantification of spatial structure of variability**

*Related to the above, in playing around with the model output (see images below), I noticed that the spatial patterns of precipitation were substantially (and unsurprisingly) noisier in the stochastic version. It struck me that the texture of the precipitation field seemed somewhat unrealistic. It would be useful to compare the simulated precipitation with observed precipitation: especially if the authors are able to run a hindcast simulation.*

*Alternatively, it might be useful to quantitatively examine the spatial statistics of precipitation: e.g., looking at structure functions (aka variograms), or equivalently power spectra. My concern*

*is that the stochastic parameterization introduces too much small-scale noise, and if so, this should be quantified and documented.*

**Reply:** Since we did not plan on conducting hindcast simulations, we analyzed the variogram using a daily averaged snapshot. As shown in Fig. R3, the stochastic parameterization in STOCH has more small-scale noise than the deterministic scheme in EAMv1 showing shorter lag distance to reach the spatial variance of precipitation over the tropics. We have documented this in the revision.

[Figure]

**Figure R3.** Semivariance of precipitation as a function of lag distance over the tropics (20ºS, 20ºN) for EAMv1 and STOCH, respectively.

***CAPE-precip relationship in observations***

*Related to the above, in Figure 9 the authors use the CAPE-precip relationship to argue why the statistics change with vertical resolution. This seemed like an ideal opportunity to compare this process-oriented relationship with observations: is the dramatic change in the CAPE-precip relationship an improvement relative? Was the bimodality in the CAPE-precip relationship in the control simulation a realistic feature (presumably not)?*

*This should be relatively easy for the authors to reproduce, given that multiple reanalysis datasets reside in a semi-public location on the DOE-funded NERSC CFS filesystem in association with the CMIP6 data that have been collected for the DOE community. If the authors are unfamiliar with how to access this repository at NERSC, their program manager can likely help direct them.*

**Reply:** On the suggestion of the reviewer, we have calculated the CAPE-total precipitation relationship using multi-year observations at the ARM Southern Great Plains and GOAmazon sites. No linear relationship is seen between total precipitation and CAPE (Fig. R4). At the SGP site, high CAPE values generally correspond to low precipitation. At the GOAmazon site, high precipitation values correspond to medium values of CAPE, somewhat resembling the stochastic simulation, although the observed CAPE values at the GOAMAzon site are much smaller than those in the simulations. Fig. R4 has been included in the revision as new Fig. 10.

[Figure]

Fig. R4: Scatterplots of total precipitation versus CAPE at the ARM (a, c & e) SGP and (b, d & f) Amazon sites for (a & b) observations calculated from multi-year sounding data (2014-2015 for Amazon and 2004-2018 for SGP), (c & d) EAMv1 and (e & f) STOCH.

**Code and data availability and appropriate use of archival repositories**

*The authors do point to the E3SM code in their "Code and data availability" section, but there are two fundamental issues with the way that the authors have done this. First, it does not appear that the code for the stochastic version of the ZM parameterization is available in the public release of E3SM, or if it is, it is not in an obvious location. The GMD policy (see (https://www.geoscientific-model-development.net/about/manuscript_types.html#item2)[https://www.geoscientific-model-development.net/about/manuscript_types.html#item2]) requires that "In the case where new code is described in the paper. . . The code should be made available."*

*The code and data policy requires that code be archived in a long-term repository, and it explicitly states that github is not appropriate for this purpose. Similar to what the authors have done with the EAMv1 data, they should upload the E3SM code–that was used for this study–and they should amend the "Code and data availability" to make it clear where the new code resides in the repository.*

**Reply:** The standard, publicly released E3SM does not have the stochastic convection parameterization in it yet, although we are currently working to include it in a future version of E3SM. Since we still have work in progress using the stochastic convection code, we prefer to defer its release to the public to a later date. Or in order to satisfy the GMD requirement, we could set up a password protected site and interested users could register and download the code. For the reviewers, if it is acceptable to GMD, we can upload a copy of the code as supplementary information for the editor and reviewers only.

**Metadata in deposited code**

*In checking out the EAMv1 data that the authors uploaded to Zenodo, I noticed that the data files lack the lat/lon arrays; this effectively renders the data files unusable unless a person knows somewhere to find the lat/lon data for this model configuration. The authors should upload new versions of the files with the lat/lon fields added.*

**Reply:** A mapping file has been uploaded to Zenodo for converting the data to that with the lat/lon arrays.

**Specific, minor issues Model version**

*I think that the title of the paper should also be amended to reflect the specific version of E3SM that was used. In browsing through the E3SM tags, I see that several iterations of the E3SMv1 model exist (e.g., v1.1.0), so the use of "E3SMv1" is ambiguous.*

**Reply:** It is v1.0. We clarify it in the title.

***Misinterpretation of O'Brien et al., (2016)***

*On lines 359–362, the authors state that the vertical resolution dependence they see is consistent with that shown in O'Brien et al., (2016). I believe this is a misinterpretation of O'Brien et al. (2016). Their resolution-dependence result is described in more detail by Rauscher et al. (2016); in Equation (2) of Rauscher et al., there is a term related to the vertical grid spacing. Once terms are rearranged to solve for W, this yields the relationship $W \propto \Delta p$. Therefore, the result of Rauscher et al., (2016) and O'Brien et al., (2016) implies that vertical velocity should increase as vertical grid spacing increases. Figure 10 seems to show higher moisture flux values for the 30L simulations, which is consistent with this theory. However, this appears to be inconsistent with the language used in the manuscript, which tangentially states that "refining horizontal resolution should result in more large-scale precipitation."*

Rauscher, S. A., T. A. O'Brien, C. Piani, E. Coppola, F. Giorgi, W. D. Collins, and P. M. Lawston, 2016: A multimodel intercomparison of resolution effects on precipitation: simulations and theory. Clim. Dyn., 47, 2205–2218, https://doi.org/10.1007/s00382-015-2959-5.

**Reply:** Thanks for pointing out the language mistakes. We rephrased the related sentences to show the relationship $W \propto \Delta p$ as you explained and cited Rauscher et al., (2016) in the revision while Fig. 10 showing the moisture flux was removed following the reviewer 2's suggestion.

***Grammar and syntax***
*There are a number of few places in the manuscript with grammar and syntax issues. I recommend that the manuscript be thoroughly proof-read by someone outside the authorship team prior to resubmitting.*

**Reply:** The grammar and syntax issues have been corrected in the revision.